
# Frequency and magnitude variability of Yalu River flooding: Numerical analyses for the last 1000 years

Hui Sheng[1], Jian Hua Gao[2*], Albert J. Kettner[3], Yong Shi[2], Chengfeng Xue[1], Ya Ping Wang[1*], and Shu Gao[1]

[1] State Key Laboratory of Estuarine and Coastal Research, School of Marine Sciences East China Normal University, Shanghai 200062, China

[2] School of Geography and Ocean Science, Ministry of Education Key Laboratory for Coast and Island Development, Nanjing University, Nanjing, China

[3] CSDMS Integration Facility, INSTAAR, University of Colorado, Boulder, CO 80309-0545, USA

**Correspondence:** J.H. Gao (jhgao@nju.edu.cn), Y.P. Wang (ypwang@nju.edu.cn).

**Abstract.** Accurate determination of past flooding characteristics is necessary to effectively predict future flood disaster risk and the dominant controls. However, understanding the role of environmental forcing on past flooding frequency and magnitude is difficult due to the deficiency of observations and too short measurement time series. Here, a numerical model HydroTrend, that generates synthetic time series of daily water discharge at a river outlet, is applied to Yalu River to: 1) reconstruct annual peak discharges over the past 1000 years and estimate flood annual exceedance probabilities; 2) identify and quantify the impacts of climate change and human activity (runoff yield induced by deforestation and dam retention) on the flooding frequency and magnitude. Climate data obtained from meteorological stations and ECHO-G climate model output, morphological characteristics (hypsometry, drainage area,

River length, slope and Lapse rate) and hydrological properties (groundwater properties, canopy

interception effects, cascade reservoirs retention effect and saturated hydraulic conductivity) are form

the significant reliable model inputs. Monitored for decades and some proxies on ancient floods allow

for accurate calibration and validation of numerical modeling.

Simulations match well present-day monitored data (1958–2012) and historical flood events literature

records (1000-1958). They indicate that flood frequencies of Yalu River increased during AD

1000-1940, followed by a decrease until the present day. Frequency trends were strongly modulated by

climate variability, particularly by intensity and frequency of rainfall events. The magnitudes of larger

floods, events with a return period of 50 to 100 years, increased by 19.1 and 13.9%, respectively, due to

climate variability over the last millennium. Anthropogenic processes were found to either enhance or

reduce flooding, depending on the type of the human activities. Deforestation increased the magnitude

of larger floods by 19.2–20.3%, but the construction of cascade reservoirs in AD 1940 significantly

reduced their magnitude by 36.7 to 41.7%. We conclude that under intensified climate change and

human activity in the future, effective river engineering should be considered, particularly for small and

medium-sized mountainous river systems, which are at higher risk of flood disasters due to their

relatively poor capacity for hydrological regulation.

## 1. Introduction

Extreme climate events have increased over the last century, threatening human life and property (Cai et

al., 2014; UNISDR, 2015; Winsemius et al., 2015). River floods are the most common and damaging of

all natural disasters globally, particularly in intensely developed river basins, deltas, and coastal regions





(Field et al., 2012; Jian et al., 2014). Globally, flood damage has led to an average annual loss of $104

billion, which is expected to increase in response to population growth and development of flood-prone

regions (Jongman et al., 2012; UNISDR, 2015).

Research has predominantly been focusing on the physical and statistical characteristics of flood

events, estimating flood probability as well as investigating flooding frequency variability in response

to urbanization, climate change, and other factors ( Sambrook Smith et al., 2010; Munoz et al., 2015;

Kettner et al., 2018; Munoz et al., 2018; Zhang et al., 2018). However, only short-term (<100 years)

fluvial gauge data exists for most rivers globally, and the existing observational data are largely affected

by human activities (Milliman and Farnsworth, 2013). These relative short records lead to large

uncertainties in the predictions of future flood disasters and is problematic in discerning whether

changes in flood frequency and magnitude are in response to climate change or human activity (Holmes

Jr and Dinicola, 2010; Yang and Yin, 2018). Determining the magnitude and frequency of historical

floods can help to predict future trends in flood disasters. To date, studies have used riverine

sedimentological records to identify the frequency and magnitude of historical floods (Gomez et al.,

1995; Paola, 2003; Munoz et al., 2018). Large floods can leave distinctive imprints in sedimentary

deposits under relatively stable sedimentary environments (Sadler, 1981; Paola, 2003). However,

sedimentary records are influenced by a range of flooding magnitudes as well as both frequent and rare

flooding events (Magilligan et al., 1998; Sambrook Smith et al., 2010). It is therefore difficult to

accurately discriminate between flood events of different scales and to quantify the frequency and

magnitude of past floods using the sedimentary record (Sambrook Smith et al., 2010). Numerical

modeling provides an alternative to observational or sedimentary records studies and can be able to

successfully reproduce basin hydrology over long term with high accuracy (Syvitski and Morehead, 1999). Consequently, in order to improve the understanding of the main controlling factors of the flooding frequency and magnitude under the impact of climate change and human activities the forward hydrological model HYDROTREND is here applied.

HYDROTREND is climate-driven hydrological water balance and transport model that simulates daily time series of water and sediment discharge as a function of climate trends and drainage basin characteristics (Syvitski et al., 1998; Kettner and Syvitski, 2008). The model creates daily water discharge at a river mouth based on a classic water balance model that consist five runoff processes: rain, snowmelt, glacial melt, groundwater discharge, and evaporation. Meteorological station data or global

circulation model output (statistics of temperature, precipitation and evaporation) and basin characteristics (basin elevation, lapse rate, equilibrium line altitude-ELA and freeze line altitude-FLA) form the input data that determine whether precipitation at a certain location will fall as rain or snow on a daily basis. The model has proven to be able to capture the range in magnitude and return intervals of peak discharge events on decadal, centennial, or longer climatic scales for small to medium-sized river

basins ($10^2$–$10^5$ km$^2$) ( Syvitski et al., 1998; Syvitski and Morehead, 1999).

    The Yalu River is a typical mountainous river that flows into a macro-tide estuary. Under the impact of large peak discharges and tidal jacking, cities of China and North Korea in the lower reaches of the Yalu River suffer severely from the flood disasters (Zhai et al., 2015). Compared with other river systems, the potential for flash flooding in mountainous rivers is susceptible to both climatic events and

human activities (Yang and Yin, 2018). Over the past 1000 years, the Yalu River witnessed a drier and cooler climatic transition during the Little Ice Age (LIA). Land reclamation, warfare, reservoir



construction, and rapid urbanization have also influenced the hydrological characteristics of the river (Sheng et al., 2019). Frequent flood disasters, drastic changes in catchment environmental and insufficient research into flooding make Yalu River become an appropriate study area for simulating,

reconstructing, and identifying how flood magnitude and frequency response to climate change and human activities over past 1000 years.

In this study, HYDROTREND is applied to numerically reconstruct and investigate the impacts of climate change and human activity (deforestation and dam retention) on the flooding frequency and magnitude for the Yalu River over the past 1000 years. Present-day (1958-2012) and long-term

(1000-1990) input climate data of the Yalu basin obtained from meteorological stations (https://data.cma.cn/) and the ECHO-G climate model, which generates synthetic time series of monthly precipitation and temperature of Yalu River over last millennium through coupled spectral atmospheric model ECHAM4 and Hamburg Ocean Primitive Equation global model (HOPE-G) (Liu et al., 2009; Liu et al., 2011). Morphological characteristics (hypsometry, drainage area, slope and latitude) and

hydrological properties (Lapse rate, groundwater properties, canopy interception effects and saturated hydraulic conductivity) are collected and processed based on guidebook of the HYDROTREND (CSDM) and previous studies (Appendix A2). The model also accepted the Yalu River's length, velocity and cascade reservoirs retention effect obtained from Wang et al., 2010 as inputs to smoothen the peak discharge at the river mouth. Except for the reliable input data, the model is calibrated by measured

peak discharge during 1958-2012 at gauging stations. The simulations of flood peak discharge of Yalu River over the last 1000 years from this calibration are then validated by historical flood events literature records including estimated flood peak flow data during 1888-1958, the number of flood

disasters in different time periods and dated flood events in past millennium (Luo, 2006). The simulated

results supported by reliable input and validation data are thus significant tools for quantifying the role

of environmental forcing on flood magnitude and frequency.

Following a brief introduction of our study site in Section 2, the research methods including model

description, source of model input data, model set up and extreme statistical method for calculating

return period of flood are depicted in Section 3. In Section 4, the model simulations first validated by

present-day field measurements obtained by Hydrological Yearbook of China and long-term flood

events (date and number of floods in different dynasties) recorded by historical flood literatures of

China. After validation, the flood frequency and values of different return intervals are next analyzed

under the impact of climate change and human activities over the last 1000 years. Next then, we

qualitatively discuss the impacts of climate change and human activity (deforestation and dam retention)

on flooding base on the wavelet analysis method, and quantitatively estimated flood frequency and

magnitude in response to basin changes using model scenarios analysis. Finally, we make conclusions

and point out the implications for the future flooding in section 5.

## 2. Regional setting

The Yalu River is located at the border between China and North Korea and originates from the

Changbai (Baekdu) Mountain. It extends 795 km south-west through steep hill slopes to flow into the

northern Yellow Sea (Chen, 1998) (Fig 1). The river contributed 90% of the total freshwater input

($25.13 \ km^3y^{-1}$) and 88% of the total sediment load ($5.18 \ Mty^{-1}$) of the total amounts that the regional

rivers contributed over the past millennium, significantly influencing the geomorphic evolution and



ecosystem of the estuarine and the adjacent coastal region (Sheng et al., 2019). The Yalu River experiences a typical temperate monsoonal climate with intense summer precipitation due to a large

inland transport of oceanic moisture during the summer monsoon (accounting for 70% of the annual rainfall). The annual mean precipitation and temperature are 863 mm and 6.2 °C, respectively. Disturbances in the upper trough of the intertropical convergence zone (ITCZ) associated with subtropical highs (typhoons and cyclones) cause intensive rainfall and flood events for the Yalu River region from July–August (Sun et al., 2011). During 1879–2002 alone, the Yalu River has flooded 51

times, including 5 large floods (affecting most of the basin), 20 local floods, and 26 more general floods depending on the flood distribution and disaster level (Luo, 2006). Most of these floods were characterized by large single-peak discharges ranging from 20,800 to 38,038 $m^3$/s typically lasting 3 days (data from Huanggou and Lishugou station in the Yalu River). Huanggou is the main hydrological station located in the lower reaches of the Yalu River, and Lishugou is located downstream of the Ai

River (the last, larger tributary of the Yalu River before flowing into the estuarine waters, in the region which experiences the highest precipitation of the basin (Fig 1).

Due to mass migration and rapid urbanization, the Yalu River region has experienced significant population growth over the last millennium from 5.2 person/$km^2$ in 1000 AD, to 10.4 person/$km^2$ in 1840, to a population density of 119.5 person/$km^2$ in 2012 (Fig 2a). Rapid population growth has altered

the regional environment due to intensified anthropogenic activity. During 1840-1985, forested areas decreased from 57.2% to 23.1% due to mass reclamation, war, and rapid urbanization. The forest cover has recently been restored to 42.6% by 2012, as a consequence of water and soil conversation measures (Fig 2b). Numerous dams have been constructed since the 1940s to minimize the threat of floods and



increase the supply of electricity. As of 2012, nine reservoirs were constructed, resulting to a total

reservoir storage capacity index (RSCI) of 93.2% (Fig 1 and Fig 2c). Shuifeng Reservoir—constructed

in 1940—is the largest reservoir of the Yalu basin and has a storage capacity of 11.6 km$^3$, contributing

44.9% to the average annual runoff (Sheng et al., 2019). The lithology and soil type are straightforward

for the Yalu River (Sheng et al., 2019). The mountains surrounding the Yalu Basin are predominantly

characterized by early Precambrian metamorphic rock and granites, including a small section of basalts

and alluvial deposits in the estuary. Brown soils dominate in the region, with the addition of muddy

dark-brown soils in the upper and middle reaches of the Yalu River.

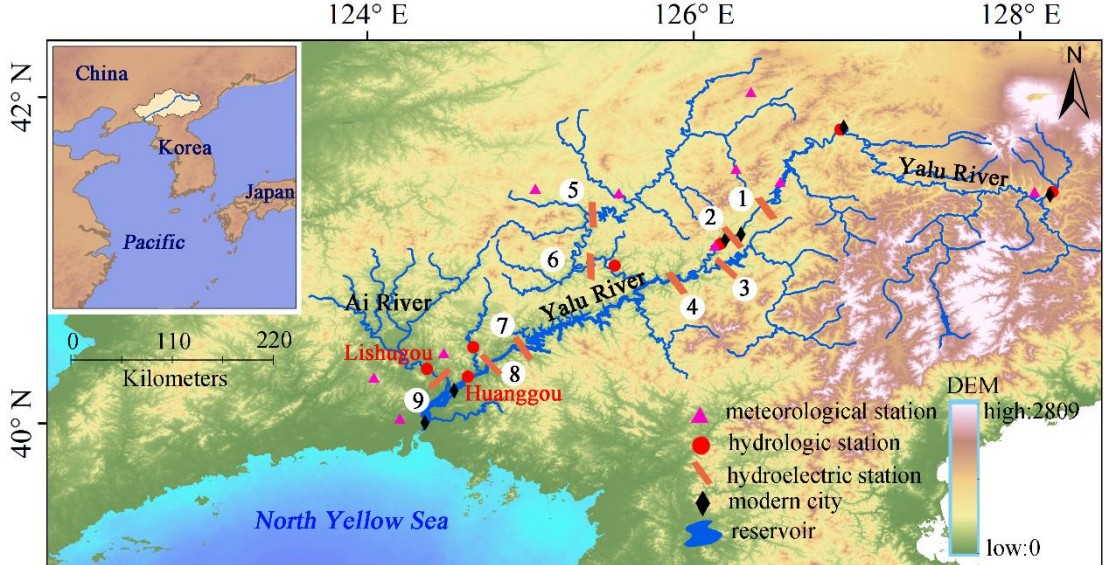

Figure 1. Map of the Yalu River basin. The total water discharge of the Yalu River is the sum of the

discharge data recorded in Huanggou and Lishuggou hydrological stations. Numbers 1 to 9 on the map

indicate the locations of the reservoirs. Digital elevation model (DEM) data is derived from ETOPO1

Global Relief Model (https://www.ngdc.noaa.gov/mgg/global/etopo1sources.html).



# 3. Method

## 3.1. Model description

The HYDROTREND hydrological model simulates daily water and sediment discharge at the river mouth and accurately predicts flood frequency and distributions (Syvitski et al., 1998). The model can simulate past ($10^0$–$10^5$ years) behavior of small and medium-sized rivers ($10^2$–$10^5$ km$^2$) by incorporating historical data on climate (meteorological data and high-resolution modeled climate data), basin properties (river networks, topography, glacier equilibrium-line) and human activity (reservoirs

and deforestation) (Syvitski et al., 1998; Kettner and Syvitski, 2008). The model has successfully estimated the long-term flux of freshwater and sediment to the coastal ocean in drainage basins across the world, including the Danube, Rhône, and Po basins in Europe (Kettner 2009, McCarney-Castle 2012), Poyang Lake (Mainland China) and the Lanyang River (Taiwan) in Asia (Syvitski et al., 2005; Gao et al., 2015), and several Greenland river systems (Overeem and Syvitski, 2010). Model

performance on flood magnitude and frequency has also been successfully tested in the flood-dominated Eel River in northern California (Syvitski and Morehead, 1999). HYDROTREND is described in detail by Kettner and Syvitski (2008) and Syvitski et al. (1998). In this study, we specifically refer to the daily water discharge methodology.

HYDROTREND simulates daily water discharge based on the classic water balance equation (Eq.1):

daily fluvial water discharge ($Q$) determined by drainage basin area ($A$) precipitation per unit area ($P$), evaporation ($E_v$) and groundwater storage and release ($S_r$). Here $ne$ is the number of simulated epochs and $i$ is the daily time step.





$$Q = A \sum_{i=1}^{ne}(P_i - E_{vi} \overset{+}{-} S_{ri}) \tag{1}$$

The climate module in HYDROTREND is used to simulate the annual precipitation and temperature

for a given epoch base. The linear trends (increasing, decreasing or constant) are used to generates

climate change scenarios and modify input mean values for given epoch.

$$T = T_s + \frac{dT_s}{dy}y + T_r \tag{2}$$

$$P = P_s + \frac{dP_s}{dy}y + P_r \tag{3}$$

Eq.2 and Eq.3 are used to predict the annual total temperature and precipitation. $T_s$, $P_s$ are the epoch

starting intercept temperature and precipitation. $\frac{dT_s}{dy}$, $\frac{dP_s}{dy}$ are epoch temperature and precipitation

change per year. $T_r$, $P_r$ are defined as normally distributed random temperature and precipitation offset.

The model input data includes the parameters in Eq.2 and Eq.3 (Appendix A2).

For a given year's total precipitation and temperature, we use the basin elevation distribution

characteristics, starting glacier equilibrium line altitude (FLA), and the temperature lapse rate to divide

the total annual precipitation into monthly rainfall ($Q_r$), snowfall ($Q_n$) and ice ($Q_{ice}$), ensuring mass

balance. Surface runoff is adjusted by groundwater evapotranspiration ($e_{gw}$) and canopy interception

($e_c$).

Monthly rainfall component ($Q_{ri}$):

$$Q_{ri} = P_i A_i \tag{4}$$

$$E_w = e_c + e_{gw} \tag{5}$$

Monthly rainfall is defined as the total precipitation for month $i$ ($P_i$) multiplied by the area of




rainfall $(A_i)$. The monthly reduction of discharge $(E_w)$ of rainfall area is based on groundwater

evapotranspiration $(e_{gw})$ and canopy interception $(e_c)$.

Monthly snowfall component $(Q_{ni})$:

$$Q_{ni} = \begin{cases} 0 & when\ h_{fl} \geq h_{ela}, "summer" & (6a) \\ P_i(t - A_i - g)(1 - E_d)(1 - x) & when\ h_{fl} < h_{ela}, "winter" & (6b) \end{cases}$$

Monthly snowfall component $(Q_{ice})$:

$$Q_{ice} = \begin{cases} P_i(t - A_i)(1 - E_d)(1 - x) & when\ h_{fl} \geq h_{ela}, "summer" & (7a) \\ P_i g(1 - E_d)(1 - x) & when\ h_{fl} < h_{ela}, "winter" & (7b) \end{cases}$$

Where $h_{fl}$ is the freezing-line altitude, $h_{ela}$ is the drainage basin elevation, $g$ is the glaciated region

above the FLA, and t is the total basin area. For the monthly snowfall $(Q_{ni})$ and ice $(Q_{ice})$

components, the discharge is decreased based on groundwater $(x)$ and evaporation $(E_d)$. Ice ablation

and snowmelt recharge to river only at an appropriate degree-day (daily temperature) condition that the

daily temperature above 5,10 and 20℃ (Syvitski and Alcott, 1995).

The random degree day module in the model is applied to generate normally daily distributed random

temperature for each month, similar to simulations of annual total temperature, the input monthly

average temperatures and their standard deviations are used to generate temperatures for each day of

month in the random degree day module (Appendix A2). The rainfall event module creates a number of

rain days for each month $(P_d)$ through the input monthly precipitation statistics data combining

Monte-Carlo technique. The output data is applied to further simulate daily water discharge induced by

the rainfall component. The amount of rainfall that reaches the ground $(P_g)$ is calculated by removing





canopy evaporation from the total daily rainfall ($P_d$). The daily surface runoff ($q_s$) is mainly determined

by saturation excess($q_{se}$), infiltration excess ($q_{ie}$), and subsurface storm flow $q_{ss}$ (from ground water

to river system).

$$q_s = q_{se} + q_{ie} + q_{ss} \qquad (8)$$

Subsurface storm flow $q_{ss}$(groundwater recharge to river) is calculated by

$$q_{ss} = \alpha_{ss} \left( \frac{GW_{store} - GW_{min}}{GW_{max} - GW_{min}} \right)^{\beta_{ss}} \qquad (9)$$

Subsurface storm flow is dominated by ground water discharge. The maximum ($GW_{max}$), minimum

($GW_{min}$), present ($GW_{store}$) groundwater storage (m$^3$), $\alpha_{ss}$(subsurface storm flow coefficient) and

$\beta_{ss}$(subsurface storm flow exponent) are given by the input file (Appendix A2).

Saturation excess ($q_{se}$) is dependent on rainfall intensity and the level of the groundwater storage pool

(GW). $\alpha_c = 0.98, \beta_c = 1.0$ are defined as saturations excess coefficient and exponent, respectively

(Sivapalan et al., 1996).

$$q_{se} = \begin{cases} 0 & when\ GW_{store} < \ GW_{min} & (10a) \\ \alpha_c \left( \dfrac{GW_{store} - GW_{min}}{GW_{max} - GW_{min}} \right)^{\beta_c} P_g & otherwise & (10b) \end{cases}$$

The infiltration excess ($q_{ie}$) is a function of the rainfall rate (reaching the ground) ($P_g$), saturation excess

($q_{se}$), and infiltration rate ($f_s$).

$$q_{ie} = \begin{cases} 0 & when\ P_g - q_{se} - f_s \leq 0 & (11a) \\ P_g - q_{se} - f_s & otherwise & (11b) \end{cases}$$

The infiltration rate ($f_s$) is calculated based on rainfall intensity ($P_g$), the level of the groundwater

storage pool (GW), saturated hydraulic conductivity ($K_0$), minimum ($P_{cr}$) and maximum ($P_{max}$)





infiltration rates, and a conversion constant (C1).

$$f_s = \begin{cases} P_g GWC1 & when\ P_g \leq P_{cr} & (12a) \\ P_g \left(\dfrac{K_0 - P_{max}}{P_{max} - P_{cr}}\right) GWC1 & when\ \ P_{cr} < P_g < P_{max} & (12b) \\ K_0 GWC1 & when\ \ P_g \geq P_{max} & (12c) \end{cases}$$

    Human land-use can also influence daily runoff at river outlets by influencing surficial soil hydraulic

properties, such as the saturated hydraulic conductivity ($K_0$), which can impact the pathway and

transmission rates of precipitation to river systems (Price et al., 2010). In this study, the $K_0\ (mm/h)$

influenced by human land-use can be expressed as follows:

$K_0 = a_1 Veg + a_2(1 - Veg)$                                                     (13)

where $a_1$ (22 mm/h in study region) and $a_2$ (3 mm/h in study region) are the saturated hydraulic

conductivities under forest and non-forest cover (Price et al., 2010), and $Veg$ is the forest coverage in

the basin.

## 3.2. Model input data

For model input we used present-day and long-term climate data of the Yalu basin (monthly averages

and standard deviations) obtained from meteorological stations during 1958-2012 (https://data.cma.cn/)

and the ECHO-G climate model output in period of 1000-1990 (Figs 2d and e). The climate model

ECHO-G, that coupled spectral atmospheric model ECHAM4 and Hamburg Ocean Primitive Equation

global model (HOPE-G), generates monthly precipitation and temperature of Yalu River over last

millennium (Liu et al., 2009; Liu et al., 2011), which is also calibrated by meteorological station data

during 1950-1990. Annual daily peak discharge data of the Yalu River (Huanggou station) and the Ai

River (Lishugou station) were obtained from the China Hydrological Statistical Yearbook (Figs 3 and 4).

We accessed soil and lithology data from the Ministry of Natural Resources of the People's Republic of

China (http://data.mlr.gov.cn/). Elevation (ASTER GDEM) and reservoir data were derived from NASA

and the National Inventory of Dams Database, respectively (Figs 1 and 2c). As shown in Figs 2a and b,

we used the millennial population and forest coverage data of Yalu basin from a recent study, which

analyzed the fluvial discharge variability of the Yalu River for the last 1000 years (Sheng et al., 2019).

Other input parameters and their sources are provided in Appendix A2.

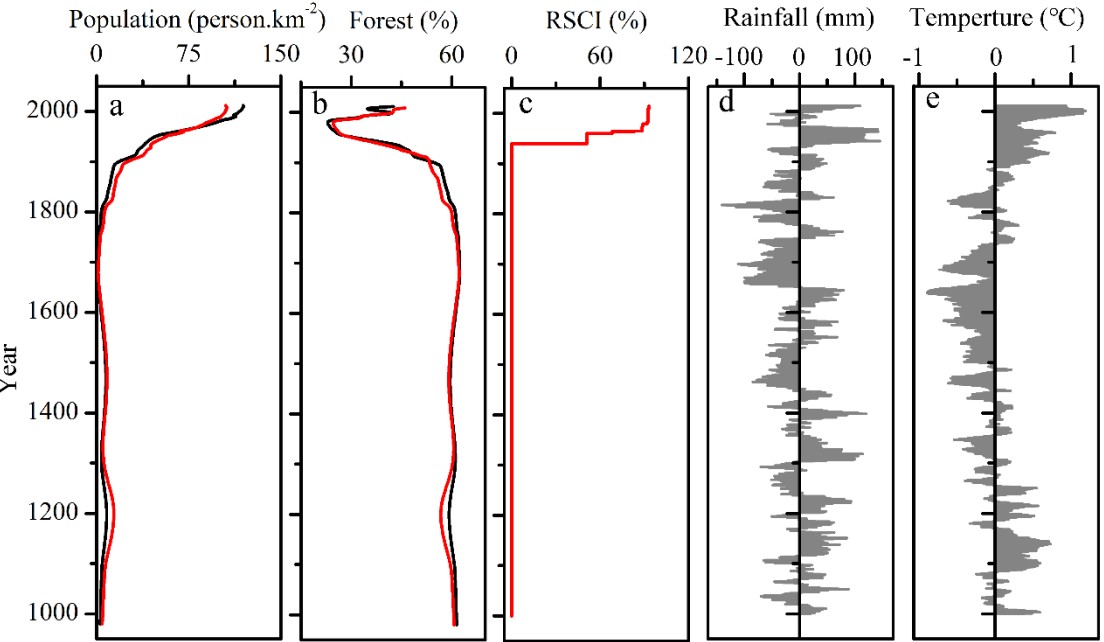

Figure 2. Model input data for the Yalu River for the period 1000–2012, including (a) population

density (Sheng et al., 2019), (b) percentage forest coverage of the basin (Sheng et al., 2019), (c) total

reservoir storage capacity index (RSCI = reservoir storage capacity/annual average water discharge); (d)

annual average rainfall anomalies, and (e) annual average temperature anomalies (Liu et al., 2009 and



2011).

## 3.3. Model set-up

Changes in monthly and daily rainfall events due to inter-annual precipitation variability strongly correlate with fluvial flood disaster occurrences (Holmes Jr and Dinicola, 2010). Initial soil conditions have varied saturation and infiltration excess capacities depending on the moisture content from previous rainfall occurrences, which determines the amount of runoff entering a river system (Sivapalan et al., 1996). For this study, we identified the periodic wet years, average years, and dry years based on multi-year precipitation data from the Yalu and Ai Rivers. Infiltration and saturation excess (groundwater storage pool) were therefore more accurately assessed based on the three different rainfall conditions. Each of the three periods (wet, average and dry years) were further divided into strong, moderate, and weak rainfall (SMW) systems (Appendix A1) to better simulate daily precipitation intensity and distribution. We used ~14 years as the period of wet and dry years for the Yalu River basin (of similar saturation excess) to simulate flooding for the past 1000 year (Yi et al., 2014). Thus, simulated daily rainfall was divided into a total of nine categories (wet year-SMW, average year-SMW, and dry year-SMW) to reconstruct the annual maximum water discharge over the last 1000 years. The model input for the rainfall event distribution coefficients and exponents were strongly correlated with the simulated daily rainfall (Appendix A2). However, we conducted a calibration analysis using partial measurements of peak water discharges (calibration period) for the Yalu and Ai rivers, as it is difficult to obtain direct measurements of these parameters in the field. Subsequently, the calibrated parameters were compared with another observed peak flow (validation period) to validate the accuracy of the

simulation (Fig 3 and 4).

Three simulation scenarios were chosen to investigate the impact of climate change and human activity on the frequency and magnitude of flooding. The first scenario is only driven by climate change (climate-Case1) over the past 1000 years (so parameters that describe the human impact were kept the same). Changes of the input parameters include annual and monthly precipitation and temperature variability, the rainfall event distribution coefficient, exponent correlation with simulated daily rainfall

values. A constant of saturated hydraulic conductivity ($15 \, mm/h$) was applied for natural conditions and the influence of dam flood retention was excluded (Appendix A2). The second scenario reflects climate change and some human impact by combining changes in climate and forest cover induced by human land-use (climate + forest-Case2). Inputs include climate data and saturated hydraulic conductivity (K0) caused by changes in forested area. The influence of dam interception was excluded.

The third scenario combines climate change, forest change, and dam emplacement for flood retention, so combines all key human impact factors as well as climate change effects (climate + forest + dam-Case3).

## 3.4 Generalized extreme value distribution

The generalized extreme-value distribution (GEV) is commonly used to estimate the highest and lowest

value among a large group of independent, identically distributed random values representing observations or simulations (Goel and De, 1993; Kim et al., 2012). The GEV combines three extreme value distribution functions (Type I - Gumbel, Type II - Fréchet, and Type III - Weibull distribution) into a single form and allows the data to decide the most appropriate distribution. The probability density





function is defined as follows:

$$H(x;\mu,\sigma,\xi) = \begin{cases} exp\left\{-\left[1+\frac{\xi(x-\mu)}{\sigma}\right]^{-\frac{1}{\xi}}\right\}, \xi \neq 0 \\ exp\{-exp[-(x-\mu)]/\sigma\}, \xi = 0 \end{cases} \quad (9)$$

where H is the GEV distribution, μ, σ, and k are the parameters for location, scale, and shape, respectively. The type of extreme value distribution is as follows determined by the shape parameter ($\xi$) of a set of random data:

(1) $\xi=0$, $H(x;\mu,\sigma,\xi)$ corresponds to Type I (Gumbel distribution) in which $x \in R$ and the tails of the distribution function decrease exponentially.

(2) $\xi>0$, $H(x;\mu,\sigma,\xi)$ corresponds to Type II (Fréchet distribution) in which $x \in [\mu+\sigma/\xi],+\infty)$ and the tail of the distribution function decrease as a polynomial.

(3) $\xi<0$, $H(x;\mu,\sigma,\xi)$ corresponds to Type III (Weibull distribution) whose $x \in (-\infty,\mu+\sigma/\xi)$ and the tails of the distribution function is finite.

GEV has been widely applied in hydrological analyses, climate statistics, and disaster reduction research (Martins and Stedinger, 2000; Kharin and Zwiers, 2005). In this paper, we used the GEV to fit the Yalu daily peak flow distribution. We calculated the flood return periods and confidence intervals to investigate the frequency and magnitude flood variability of the Yalu River under the impact of climate change and human activity.

## 4. Results and discussion

### 4.1 Model validation





### 4.1.1 Present-day flood validation

To validate the model and calibrate its input parameters, we used the annual maximum peak flows at two gauging stations for 1958–2012 (the Yalu River data consists of data from the Yalu-Huanggou mean

river and its downstream tributary the Ai-Lishugou; Fig. 1), accessed from the Hydrological Statistical Yearbook of the Heilongjiang basin. As shown in figures 3d and 4d, the climate-driven model adequately captures the variability in peak discharge measured at the gauging stations. The model output is limited by uncertainties in the data on climate (rainfall and temperature) and human activity (deforestation and dams). However, the peak flow ranking data between model output and observations

show a similar trend, inferring adequate model performance. HYDROTREND closely simulates the observed peak flow distribution as well as the maximum and minimum discharge during wet, average, and dry years (Figs 3e–g and Figs 4e–g). For this study, different return interval flood values were calculated using the GEV statistical method based on the gauged and simulated daily maximum runoff data of the Yalu River basin from 1958 to 2012. Results show that the relative error of the daily peak

flows between the simulation and observations were <10.6% for all return intervals (Table 1). We therefore confirm that the model can accurately capture flood magnitudes and recurrence intervals for the Yalu River.



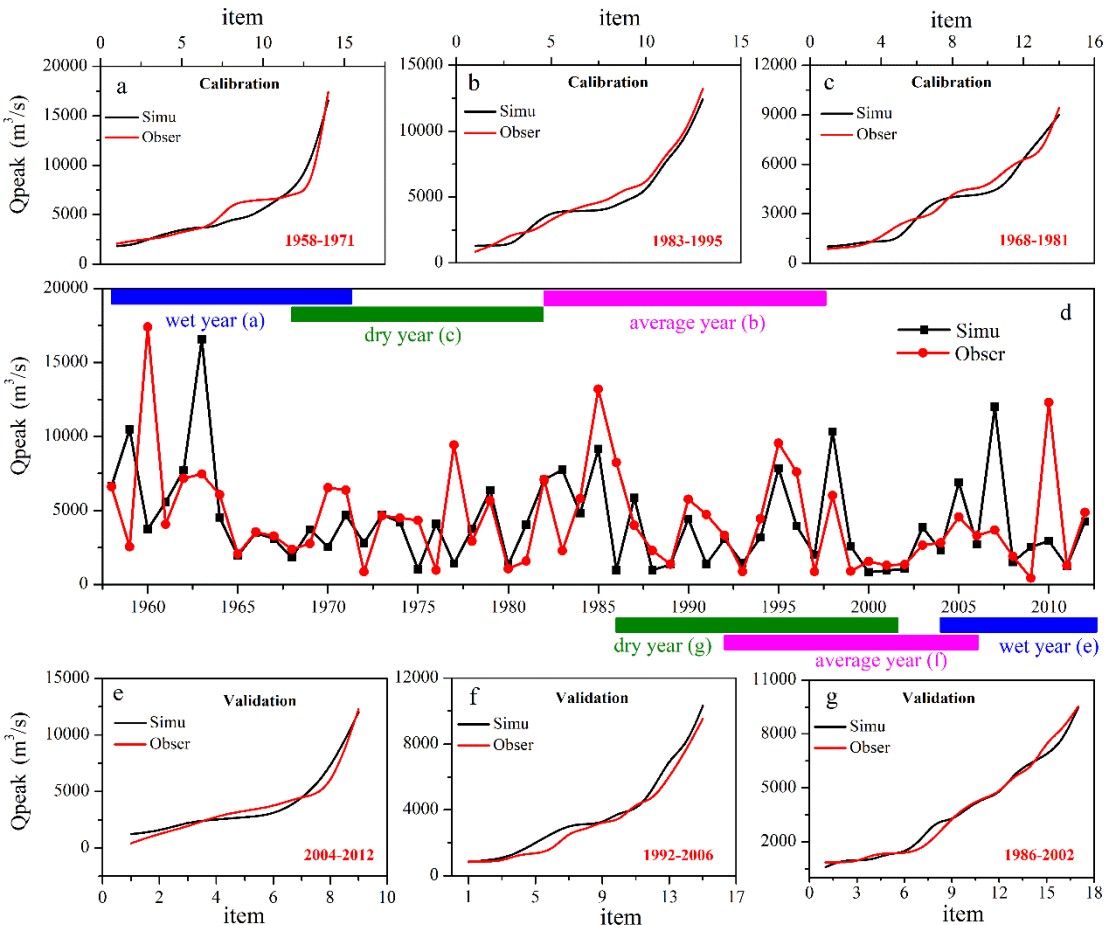

Figure 3. Comparisons of simulated and observed peak discharge of the Ai River (Yalu River tributary):

a, b, and c show ranked peak flows for model simulations and observations for wet, average, and dry

years during the calibration period, respectively; e, f, and g show ranked peak flows between the model

simulations and observations for wet, average and dry years during the validation period, respectively;

and d is the time-series comparison of simulated and observed daily peak flow from 1958–2012.





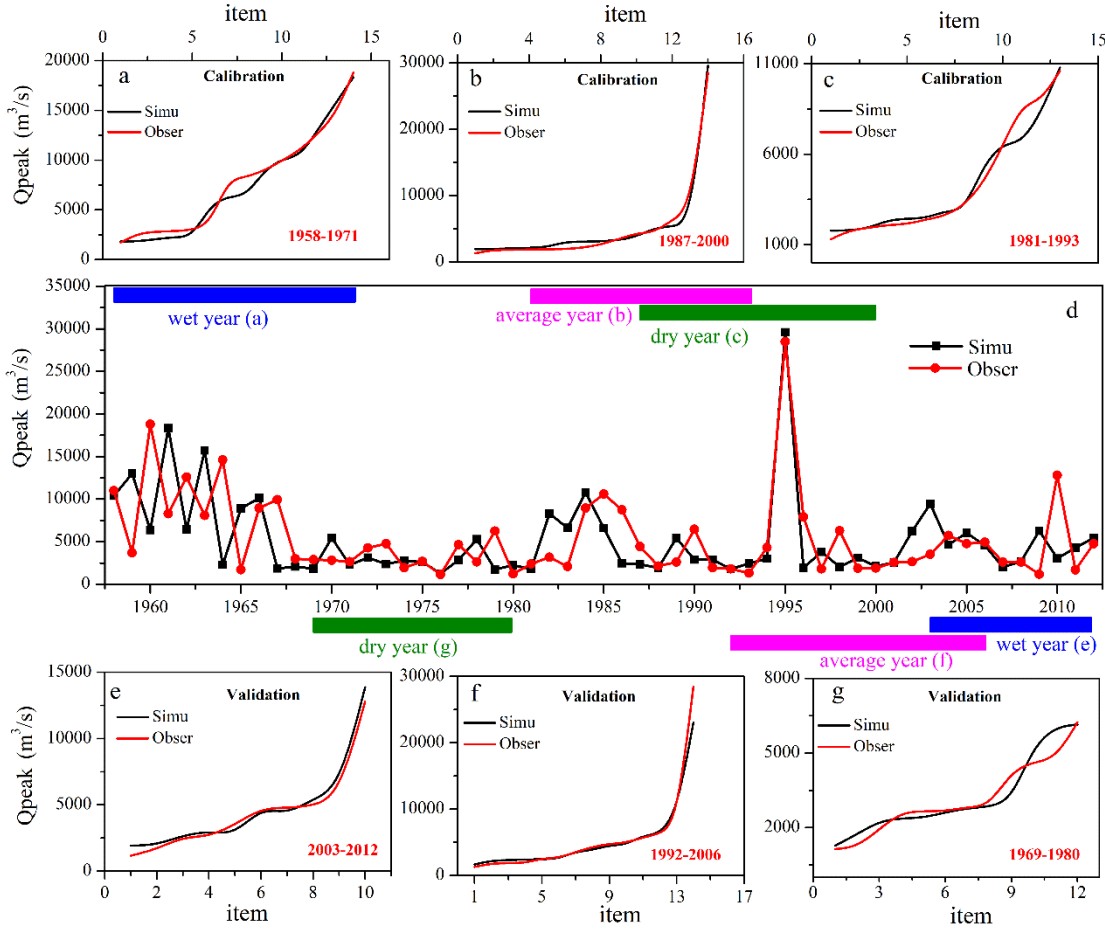

Figure 4. Comparisons of simulated and observed peak discharge of the Yalu River: a, b, and c show ranked peak flows between the model simulations and observations for wet, average, and dry years during the calibration period, respectively; e, f, and g show ranked peak flows between the model simulations and observations for wet, average, and dry years during the validation period, respectively; and d is the time-series comparison of simulated and observed daily peak flow from 1958–2012.



Table 1. Comparison between observed and simulated return interval peak discharges for the period 1958–2012.

| Flood return intervals | 200 year | 100 year | 50 year | 20 year | 10 year | 5 year | 2 year |
|---|---|---|---|---|---|---|---|
| Simulation | 71,923 | 52,184 | 37,781 | 24,500 | 17,490 | 12,267 | 7,108 |
| Observation | 71,842 | 54,116 | 40,473 | 27,094 | 19,564 | 13,628 | 7,339 |
| Error (%) | 0.1% | -3.5% | -6.7% | -9.6% | -10.6% | -9.9% | -3.1% |

The unit of simulated and observed peak flow is m$^3$/s.

Percentage error is estimated as follows: (simulation-observation)/observation.

### 4.1.2 Validation of long-term flood events

We used historical flood records of the Yalu River over the past 1000 years to further verify model performance. Estimates of peak flow data of the Yalu River from 1888–1958 and historical data of flooding disasters from 1000–1888 were obtained from the "Compilation of historical flood survey data in China" (Luo, 2006). The peak discharge observed in 1923 (32,000 m$^3$/s) and 1907 (20,800 m$^3$/s) were used to define the Yalu River's "devastating floods" and "immense floods" respectively, based on

historical flood records (these include whole basin large flooding and local large flooding of the Yalu River) and estimated peak flow data from 1888-1948 (Fig 5). Records of historical floods for the Yalu River are relatively scarce during 1000–1234, and flood events that have been adequately dated are predominantly "devastating floods" occurring between 1235–1888. However, historical records also identify the number of lower magnitudes "immense floods" that occurred between 1251–1368 (the Yuan

dynasty in China), 1369–1638 (the Ming dynasty in China), and 1791–1910 (Late Qing Dynasty in China).





Validated results indicate that the occurrence frequency of devastating floods estimated by using the simulated peak flows matched the historical records; we identified high frequencies of devastating floods during 1250–1350 and 1840–1950, and a lower frequency of devastating floods during 1400–

1800 (Fig 5). In contrast, the number and frequency of immense floods were similar for all time periods. Recorded historical frequencies of immense floods of 22.0% and 20.8% were similar to the model simulated frequencies of 21.2% and 18.4% during 1251–1368 and 1911–1958 (periods of higher rainfall intensity), respectively (Table 2). In contrast, due to lower precipitation intensities during the periods 1369–1638 and 1791–1910, the recorded historical frequencies of immense floods reduced to 11.9%

and 10.8% relative to 13.0% and 10.0% based on the model simulations (Table 2). These results confirm accurate model simulations of long-term flooding variability for the Yalu River basin.

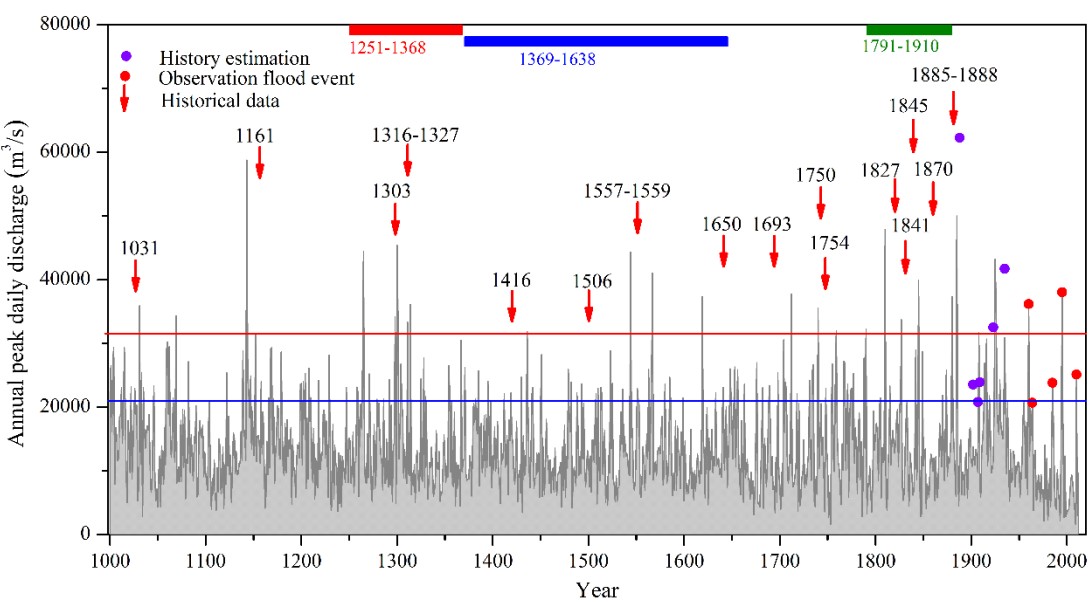

Figure 5. Historical flood records and model simulated annual peak daily discharges for the Yalu River over the past 1000 years. Red arrows indicate adequately dated historical records of devastating floods;





the red and blue line indicate the minimum peak discharge threshold to define devastating and immense

floods, respectively; The red, blue, and green columns indicate the time periods in which sufficient data

of the number of immense floods were available.

Table 2. The number and frequency of flood disasters for different time periods based on historical data

and model simulations

| Statistical periods | Number of flood disasters | | Flood occurring frequency | | Qpeak discharge ($m^3/s$) |
|---|---|---|---|---|---|
| | Reconstructed observations | Simulation | Reconstructed observations | Simulation | |
| 1251-1368 | 26 | 25 | 22.0% | 21.2% | 20,800 |
| 1369-1644 | 32 | 35 | 11.9% | 13.0% | 20,800 |
| 1791-1910 | 13 | 12 | 10.8% | 10.0% | 20,800 |
| 1911-1958 | 10 | 9 | 20.8% | 18.4% | 20,800 |

Qpeak discharge: the minimal flood value to determine occurrence of a flood event

## 4.2 Flood frequency analysis over the past millennium

### 4.2.1 Flood value estimates of different return intervals

River flood return intervals are estimated based on annual peak discharges. The accuracy of flood

frequency estimations improve with longer timescales of peak flow data (Holmes Jr and Dinicola, 2010).

Currently, most rivers globally have <100 years of fluvial gauged data, which can be used to accurately

estimate at least 100-year flood return intervals (Milliman and Farnsworth, 2013). However, one has to

be cautious when applying these relative short datasets to estimate longer-term flood return periods

of >500 years, as uncertainties rapidly increase by extrapolating return periods beyond the time period



of observations. For this study, we were able to estimate higher return interval floods by combining the

past 1000-year model simulated annual peak discharges of the Yalu River basin with the GEV statistical

analysis (Fig 6). The statistical analysis show that the peak flows for the 10,000-year return flood event

for the Yalu River is 88,321 m³/s. Peak discharges for the 1000-year and 100-year return interval floods

were 61,388 and 40,080 m³/s, respectively (Fig 6).

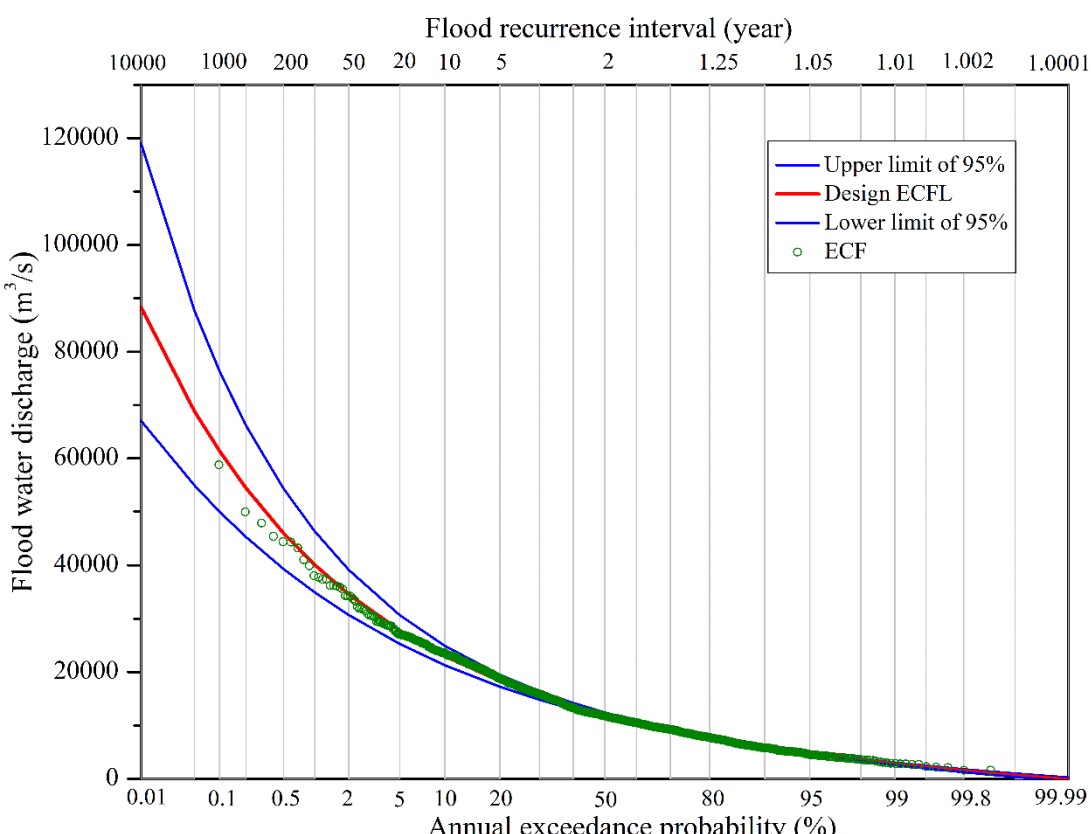

Figure 6. Fitted frequency curves of annual maximum daily discharge for the Yalu River based on the

GEV statistical method and simulated annual peak flows of the past 1000 years. The blue lines indicate

the upper and lower limits of the 95% confidence level, the red line indicate that design empirical

cumulative frequency line (ECFL) and the green dots are the empirical cumulative frequency (ECF) for





annual peak water discharges over the past millennium.

## 4.2.2 Flooding return intervals over the past millennium

Studies have indicated that the return intervals of river flooding adjust in response to climate change and human activity (Milly et al., 2002; Milly et al., 2005). Altered rainfall patterns (frequency, intensity and spatial distribution) caused by climate variability and the influence of human activity (land use, impoundment, or diversion) on river runoff have significantly altered flood return periods (Holmes Jr and Dinicola, 2010; Price et al., 2010). Both the climate and human activity for the Yalu River basin

have changed dramatically over the past 1000 years. The climate of the Yalu River basin was colder and drier during 1451−1840: a period known as the Little Ice Age (LIA) (Paulsen et al., 2003). During the LIA, the annual average rainfall and temperature in the region was 793 mm and 4.85 ℃, respectively; the annual average precipitation reduced by 18 mm and 21 mm, and the annual average temperature decreased by 0.55 ℃ and 1.0 ℃ relative to the periods 1000−1450 and 1841−2012, respectively (Fig 2d

and e). Discharge of the Yalu River fluctuated between 6.4%−11.4% under the influence of climate change (Sheng et al., 2019). In contrast to multi-year annual average precipitation, the frequency of extreme precipitation events for the Yalu River showed little difference between 1451−1850 and 1000− 1450, 5.90% and 6.67%, respectively. However, the frequency of extreme rainfall events sharply increased to 10.47% during 1840−2012 in response to changes in climate and human activity (Fig 2).

During 1000−1840, the basin had a population density of only 5.27person/km$^2$ and ~60% of the basin was covered by forest (Fig 2a and b). However, immigration, land reclamation, war, and rapid urbanization reduced forest coverage from 55% in 1840 to 30% in 1940 (Fig 2b). Further, the





construction of the dam in 1940 significantly influenced the hydrological characteristics of the Yalu River (Fig 2c).

Flood return intervals of the Yalu River over the past 1000 years first show an increasing trend during 1000–1941, followed till today by a decrease in response to climate change and human activity (Fig 7). Higher precipitation was observed during 1000–1450 (816.5 mm/year) relative to 1450–1840 (793 mm/year; LIA), but the intensity and frequency of extreme rainfall events were similar between the two periods. Climate change led to a 5.4% decrease in flood magnitude for the different flood return

intervals during the LIA relative to the period 1000–1450. The average annual rainfall for the basin during the period 1841–1940 was similar to the LIA (1450–1840), but the intensity and frequency of extreme rainfall was significantly higher during 1841–1940 (8.0%) relative to the LIA (5.90%) (Liu et al., 2009; Liu et al., 2011). The estimated peak discharge of the different flood return events significantly increased during 1841–1940, and climate change had a greater impact on the 100-year and

50-year floods relative to the shorter-term return events (Fig 7). The estimated peak discharge of the 100-year and 50-year return floods during 1841–1940 increased by 16.4–18.0% compared with the LIA, and the 20-year, 10-year, and 5-year recurrence events increased by 11.7–14.4% due to the increase of the frequency of extreme rainfall events.

Higher peak discharges of the different flood recurrence events during AD 1841–1940 can be

predominantly attributed to the increase in the intensity and frequency of extreme rainfall events. However, anthropogenic influences including immigration, reclamation, war, and urbanization in the basin also contributed to the observed increase in the peak discharges. The Yalu River basin experienced higher rainfall intensity and increased human land-use coverage during 1941–2012 relative to 1841–



1940, but the flood peak discharge had significantly reduced due to the construction of cascading

reservoirs. Following the construction of the dam in 1940, estimated peak flows for the 20-year, 10-year, and 5-year return events decreased by 16.8–23.6%, and the 100-year and 50-year recurrence intervals decreased by 9.9–12.8%.

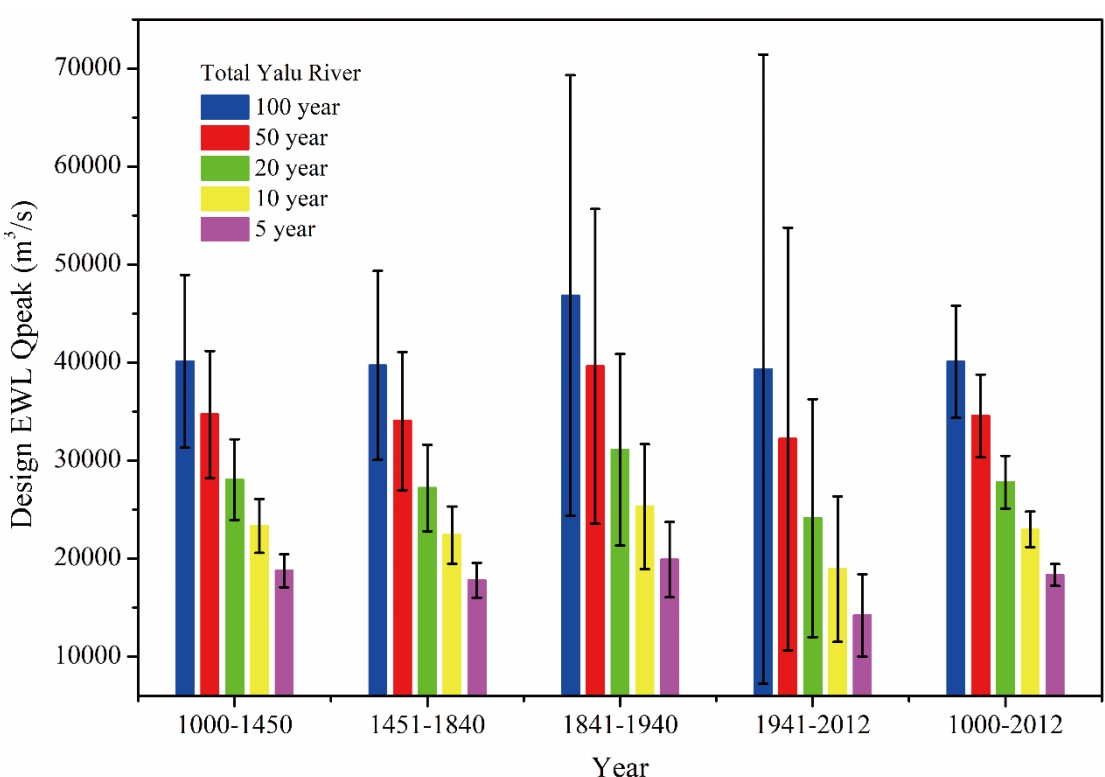

Figure 7. Estimated peak discharges of the different flood recurrence intervals for the Yalu River based

on simulated peak discharges during five periods combined with the GEV statistical method.

## 4.3 Factors controlling flood frequency variability

We conducted a wavelet analysis on flood occurrence for the different flood return intervals during 1000–2012 to further investigate the dominant controls on flood frequency variability for the Yalu River





over the past 1000 years (Fig 8). The peak discharges of the different flood magnitudes were also used

to calculate flood frequencies for the Yalu River (Fig 6 and Table 3). The wavelet results show that

during 1130–1190, 1280–1340,1520–1580, and 1880–1940, the frequencies of larger floods (such as

the 100-year, 50-year, and 20 year events) were much higher than that of other periods due to extreme

precipitation events (Fig 8). The frequencies of the 50–year events in AD 1000–1450 and 1451–1840

were 1.33% and 1.35%, respectively, as both periods experienced similar extreme precipitation events,

and during 1841-1940 rapidly increased to 4.95% in response to increasing rainfall events (Table 3).

Our results demonstrate that the frequency and intensity of extreme precipitation caused by climate

change have a dominant control on the frequencies of large floods. However, medium and

small-magnitude floods (10-year and 5-year) are more closely linked to long-term climatic trends of

warming and humidity (Fig 2 and Fig 8). Highest frequencies for the 5-year return events were observed

during 1000–1450 at 22.84% relative to 21.48% during the LIA and 21.78% during 1841–1940 due to

higher annual average precipitation for the Yalu River basin (Table 3).

As shown by Fig 8, the frequencies of the different return interval floods rapidly decrease after 1940

due to the construction of cascading reservoirs, despite the increasing frequency and intensity of

extreme precipitation events in response to climate change and anthropogenic impacts. The flood

frequencies of the 50-year and 5-year events in 1941–2012 decreased by 44.7% and 56.0%,

respectively, relative to 1841–1940 due to the construction of flood retention dams (Table 3). The

results show that the construction of reservoirs can effectively reduce flood disasters for the Yalu River

basin despite having little effect on the long-term runoff to the sea (Sheng et al., 2019).



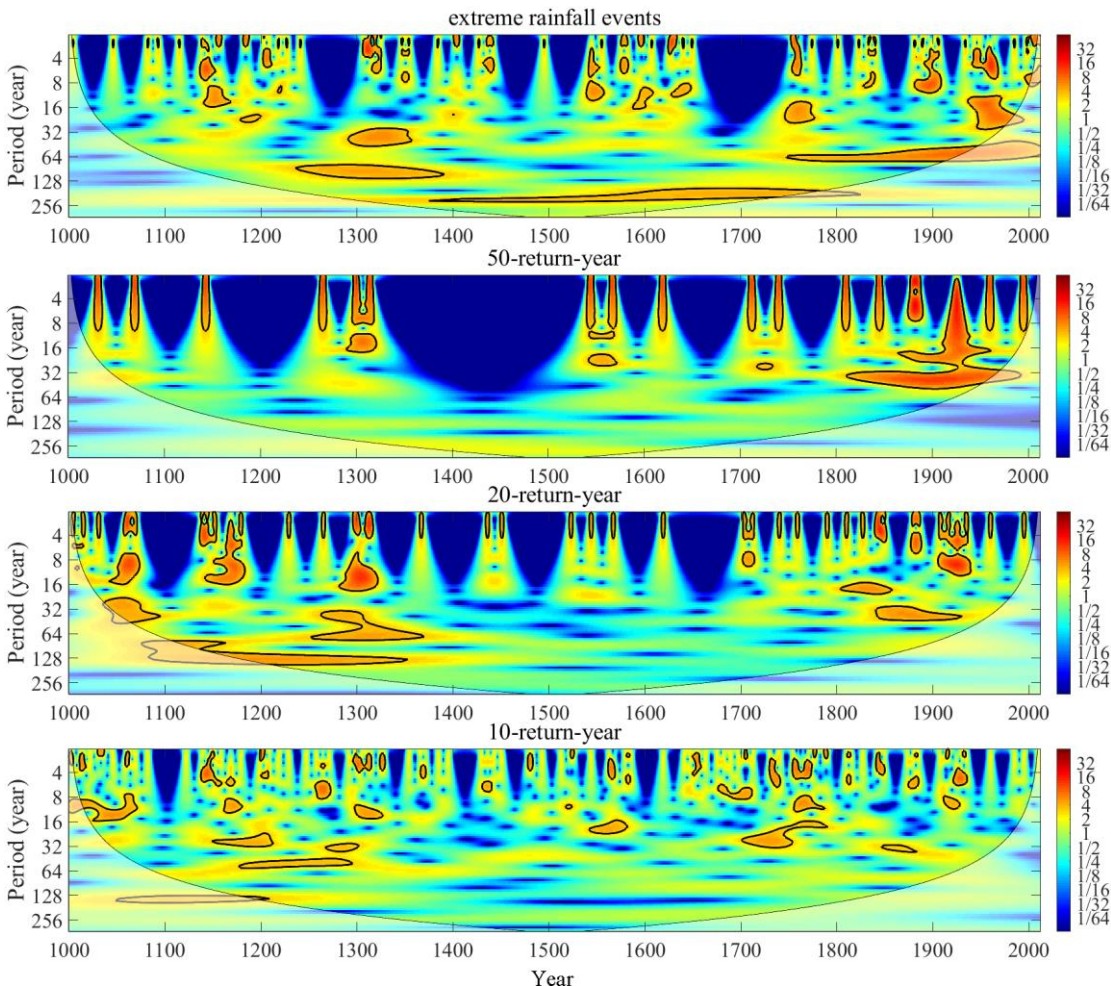

Figure 8. Wavelet analyses of extreme rainfall events and the 50-year, 20-year, and 10-year return flood events during AD 1000–2012.





Table 3. Flood frequencies for return intervals for different periods

| Period (year) | Frequency of flood occurrence for different recurrence intervals | | | | |
|---|---|---|---|---|---|
| | 100-year | 50-year | 20-year | 10-year | 5-year |
| 1000–1450 | 0.67% | 1.33% | 4.88% | 11.97% | 22.84% |
| 1451–1840 | 0.77% | 1.53% | 3.07% | 11.00% | 21.48% |
| 1841–1940 | 1.98% | 4.95% | 9.90% | 12.87% | 21.78% |
| 1941–2012 | 0.00% | 2.74% | 2.74% | 5.48% | 9.59% |

## 4.4 Flood simulation under climate change and human activity

### 4.4.1 Quantitative estimation of flood values of different return interval floods in response to basin change

To quantify the impact of climate change and anthropogenic activities on basin floods, we set up three different scenarios: Case1, climate change alone; Case2, climate change + forest cover change; and Case3, climate change + forest cover change + emplacement of dams for flood retention. Although the climate during 1000–1450 was warmer and wetter than that of the LIA, the fitted flood frequency curves of the two periods were similar when driven only by climate change (Case1). However, the flood frequency curves of 1841–2012 are significantly higher than the other two periods (1000-1450 and 1451–1840) due to the higher frequency of extreme rainfall events (Fig 9a). These results further confirm that flood frequency for the Yalu River is controlled by the frequency and intensity of extreme rainfall. The frequency of the 100-year flood recurrence interval for the Yalu River basin during 1000–1840 increased to a 50-year recurrence interval during 1841–2012 under the influence of climate change. Further, the estimated flood magnitude of the 100-year, 50-year, and 20-year floods for 1841–2012





increased by 19.1%, 13.9%, and 7.77% respectively compared to 1451–1840 (Fig 9a).

Human activities only started to significantly influence the Yalu River basin since 1840, and thus we

475 only compared the flood return intervals of the three scenarios (Case1–3) during 1841–2012 (Fig 9b).

When comparing the fitted flood frequency curves of Case2 with Case1, we found that the reduction of

forested area (conversion of forested area to agricultural land) for the Yalu basin increased the

likelihood of floods (Fig 9b). Under the impact of human land-use, the flood magnitude of the 100- and

50-year events increased by 19.2–20.3%, while the 20-, 10- and 5-year events increased by 22.0–26.3%.

480 Human land-use increased the frequency of the 20- and 10-year floods to 10- and 5-year floods,

respectively, which therefore significantly increased the occurrence likelihood of small and

medium-sized floods in the Yalu basin (Fig 9b).

The simulated scenarios for Case2 and Case3 infer the significant reduction in the frequency of flood

occurrence due to the construction of the cascading reservoirs: the return frequency of the 20-year flood

485 had decreased to a return period around 50 or 100 years; the return frequency of the 10-year flood

decreased to a 20–50 years return period; and the flood magnitude of the 100-, 50-, 20- and 10-year

events rapidly decreased by 36.7–41.7% (Fig 9b). Although the dams, build for flood retention, have

significantly reduced the magnitude of floods for the Yalu basin, the flood magnitude of the different

recurrence intervals during AD 1841–2012 were still higher compared to the period 1000–1840 due to

490 the increase of extreme climate events. Future flooding of the Yalu River basin could therefore even

more increase.



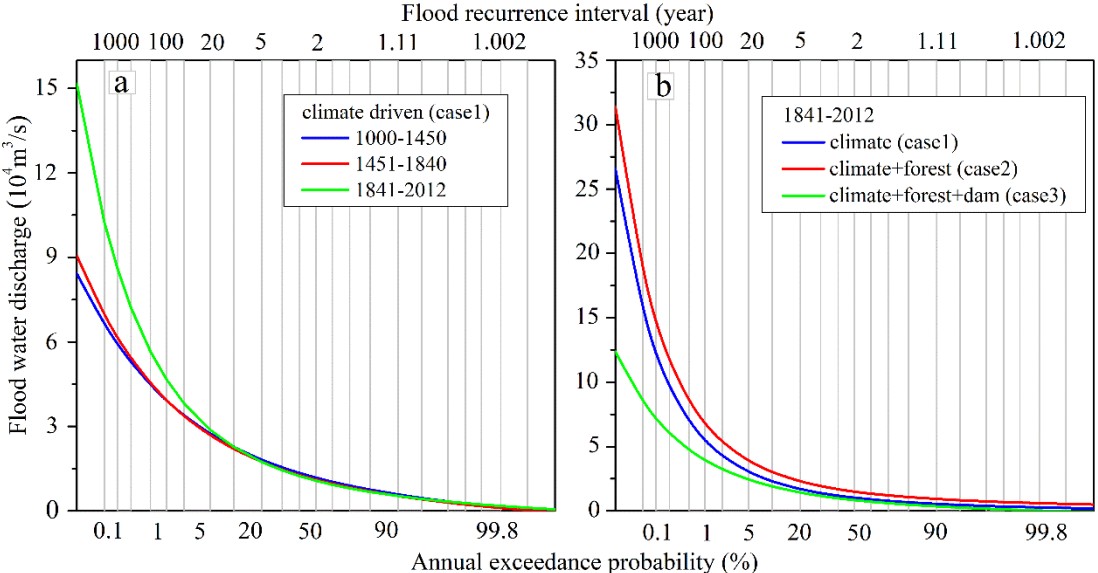

Figure 9. Frequency curves of annual maximum daily discharge for the Yalu River under the different scenarios: (a) frequency curves for the different periods only driven by climate change; and (b) frequency curves for the three scenarios (climate, climate + forest cover, and climate +forest cover + dam) during 1841–2012.

## 4.4.2 Quantitative flood frequency predictions for basin changes.

To quantify the impact of individual factors (such as climate change, forest cover, and dams) on flood frequency for the Yalu River, we calculated the flood frequencies of the different periods based on three scenarios and estimated flood magnitudes for the different flood return intervals (Fig 10). For Case1, the flood frequency of the 50-year event increased 3.8 times from 1.1% during 1000–1450 to 4.2% during 1941–2012. In contrast, the flood frequency of the 10-year event only increased by 3.4% and fluctuated between 9.1%–12.5% (Fig 10). These results further confirm climate change to be a significant driver of flood variability in the region, with higher magnitude floods being more sensitive to climate change

relative to medium and small flood events.

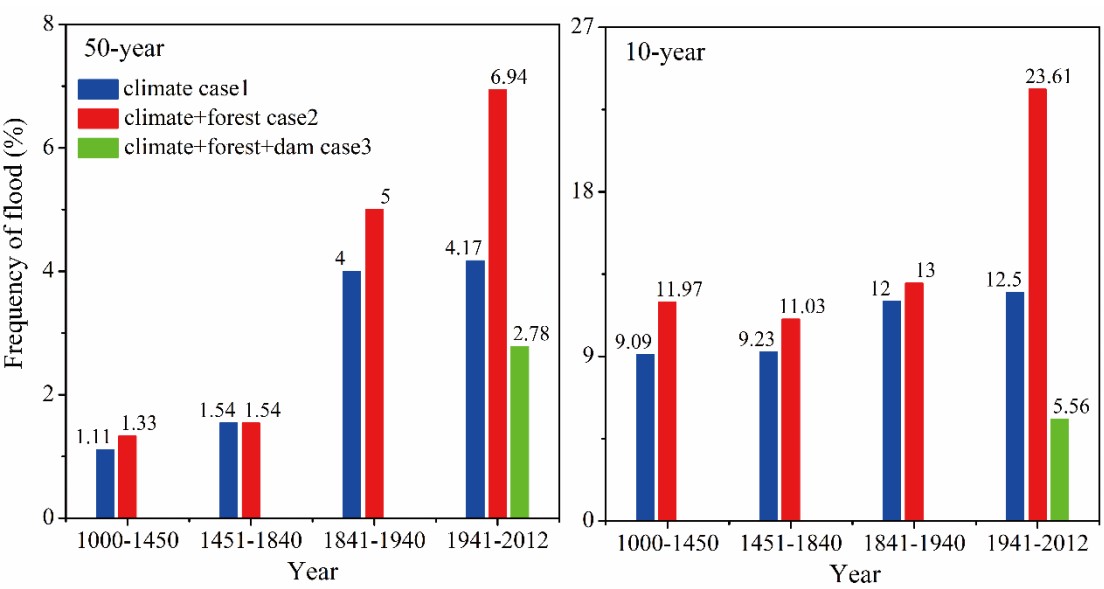

Figure 10. The flood occurrence frequencies of the 50- and 10-year events for the river basin during AD

1000–2012.

In addition to climate change, anthropogenic activities significantly influenced flood frequency (Fig

10). Deforestation as a result of human land-use increased the flood frequency for the Yalu River. When

comparing the flood frequencies for Case1 and Case2 during 1000–1840, the frequencies of floods

induced by human land-use increased by 9.1%–26.0%. Flood frequency significantly increased even

more (42.9%–53.3%) with increasing land–use for the period 1941–2012. In contrast to the impact of

land-use, the construction of dams considerably decreased the flood frequency of the Yalu River. In the

Case3 scenario, the frequency of the 50-year flood is 2.78%, which was significantly lower than the

Case2 scenario, which is 6.94%. However, the frequency was still higher than the period 1000–1840

(1.33%). During 1940-2012, the frequency of the 10-year flood reduced to 5.56%, which was its lowest

observed throughout the entire 1000year record. Dam construction had decreased the frequencies of the 50-year and 10-year floods by 4.17% and 18.05%, respectively, accounting for 59.8% and 76.5% of the

50-year and 10-year flood frequencies driven by the Case2 scenario.

## 4.5 Future flooding implications

Both observational data and model projections point towards increasing intensity and frequency of extreme precipitation events worldwide with some regional variability (Jian et al., 2014). In general, the impacts of global warming on the distribution of energy and the water-atmosphere cycle are increasing

the frequency of extreme precipitation events. Coupled climate and hydrological models have also projected an increase in extreme floods in future (Dankers and Feyen, 2008; Hirabayashi et al., 2013; Alfieri et al., 2015). In addition to climate change, human activities such as river engineering (flood diversion, dam construction, and water storage) and land-use change (agricultural and urbanization) will directly or indirectly affect the intensity and frequency of fluvial flooding (Willett et al., 2007; Price et

al., 2010; Jian et al., 2014). River basin conditions will determine the discharge characteristics, what percentage of rainfall will be routed as (sub)surface runoff, which will be amplified by deforestation, increasing the magnitude and frequency of flood events. In contrast, river engineering including flood diversions, dam construction, and water storage will reduce the chance of flooding.

Increasing forest coverage can minimize the magnitude and frequency of future extreme floods to a

certain extent. However, without the implementation of adequate water conservancy measures, the risk of flood disasters will increase in response to increasing intensity and frequency of extreme rainfall events. Furthermore, the risk of flood disasters in small to medium-sized river basins is more significant


compared to larger rivers. As larger rivers with abundant tributaries and lakes have a larger buffering

capacity to temporary store access water and therefore prevent flooding under high-intensity rainfall

events. In contrast, small and medium-sized rivers are more sensitive to extreme rainfall events, and

localized extreme precipitation events caused by tropical storms and cyclones are more likely to cause

extreme flooding.

## 5. Conclusions

The hydrological model HYDROTREND coupled with the high-resolution climate model ECHO-G

successfully captured the magnitude and frequency of flood events for the Yalu River over the last 1000

years. Over this period, flood frequencies had initially increased during 1000-1940, followed by a

decrease to the present day. The frequencies of the higher magnitude 100-year and 50-year return floods

significantly decreased for the Yalu River over the last century, but remained higher than during AD

1000–1840. Furthermore, the 20-, 10- and 5-year flood frequencies were the lowest over the last century.

The larger magnitude floods are predominantly controlled by the intensity and frequency of extreme

rainfall events, while the medium and small magnitude floods were predominantly linked to long-term

cycles in temperature and humidity. The frequencies of the 50-year and 10-year flood events for the

Yalu River increased by 210% and 33.3%, respectively, under the impact of climate change since 1840.

Unlike climate change, we found human activity to either enhance or reduce flood disasters in the

region depending on the type of activity. Flood frequencies for the Yalu River have increased by 26.0%−

53.3% due to an increase in human land-use during 1840–2012, while the construction of cascading

reservoirs effectively reduced flooding after 1940. Dam interception has significantly reduced the



frequencies of the different magnitude floods by 59.8–76.5%.

The case from the Yalu River indicates that, compared with larger basins, mountainous rivers are

more prone to flood disasters due to their relatively poor capacity for hydrological regulation when

responding to extreme climatic events. Therefore, the implementations of effective river engineering

measures (such as flood diversions and dam construction) are necessary to minimize flood risks.

Furthermore, the current flood prevention standard should also be revised due to the increasing

frequency and magnitude of flooding in the region. The use of HYDROTREND coupled with climate

model predictions to quantify flood magnitudes and frequencies are essential, but further studies are

needed to address the uncertainty in the data for climate change predictions and to better understand

various complex influencing factors in flood simulation.

*Code and data availability.* The modeling code is available on CSDMS (https://csdms.colorado.edu

/wiki/Model:HydroTrend) and The data is available upon request.

*Acknowledgement.* This research was supported by the Natural Science Foundation of China (Grant

Nos. 41576043, 41625021 and 41530962) and the Innovation Program of Shanghai Municipal

Education Commission (2019-01-07-00-05-E00027). AJK was supported through the US National

Science Foundation, Grant No. 0621695.

*Competing interests.* The authors declare that they have no conflict of interest.

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





**Appendix**

A1. Different rainfall conditions (wet, average and dry years) and rainfall forms (strong, moderate and

weak) for the Yalu and Ai rivers

| Rainfall condition | Average Rainfall (mm/year) | | Rainfall intensity | Yalu (mm) | Ai (mm) |
|---|---|---|---|---|---|
| | Yalu River | Ai River | | | |
| | | | S | >942 | >1197 |
| Wet-year | >897 | >1035 | M | 788-851 | 956-1197 |
| | | | W | <788 | <956 |
| | | | S | >939 | >1092 |
| Average-year | 820-897 | 939-1035 | M | 761-939 | 850-1092 |
| | | | W | <761 | <850 |
| | | | S | >926 | >1040 |
| Dry-year | >820 | <939 | M | 751-926 | 807-1040 |
| | | | W | <751 | <807 |

S, M and W are defined as strong, moderate and weak rainfall (SMW) forms in

different rainfall conditions (wet, average and dry years).

A2. Most significant input parameters of HYDROTREND: an example for Yalu River

| Input parameters | Source | Example: Yalu River |
|---|---|---|
| Start year; epoch; step-length: D M S or Y | User-specific | 1938; 14; D |
| Temp: start (℃); change (℃ yr$^{-1}$); STD (℃) | Meteorological data; | 5.68; -0.01; 0.75 |
| Precip: start (m); change (m yr$^{-1}$); STD (m) | Liu et al. (2009, 2011) | 0.9054; 0.0087; 0.1107 |
| Mass bal.coef; rainfall event distribution coef, distribution range | Calibration based on Hydrological data | 1.4; 1.2; 1.6 |
| Base flow (m$^3$ s$^{-1}$) | Hydrological Yearbook | 615 |
| Climate variable: monthly mean Temp (℃); within month Std Dev. of T; monthly mean Precip (mm); Std | Meteorological data Liu et al. (2009, 2011) | |





| Dev of the monthly P: | | |
|---|---|---|
| January | | -12.26; 2.17; 15.68; 14.9 |
| April | | 5.19; 1.71; 47.44; 24.3 |
| August (similar for other months) | | 20.47; 1.36; 167.07; 80.92 |
| Lapse rate ($^{\circ}C\ km^{-1}$) | http://www.the weather prediction.com | 6.0 |
| Glacier equilibrium line altitude (m), change (m $yr^{-1}$) | Meteorological data; Kezhen Zhu 1972 | 3500; 0 |
| Dry precip (nival and ice) evaporation fraction | Meteorological data | 0.65 |
| Canopy interception alphag(mm/d); betag | Sivapalan et al., 1996 | -0.1;0.85 |
| River - length (km) | Calculated (GIS) | 719.3 |
| River mouth velocity coef (k) and exp (m) | Hydrological Yearbook | 0.6203; 0.0090 |
| Initial groundwater storage ($m^3$) | Ministry of Natural | $7.38e^{+9}$ |
| Maximum and minimum groundwater storage ($m^3$) | Resources of People's Republic of China | $1.44e^{+10}$; $3.28e^{+9}$ |
| Groundwater (subsurface storm flow) coef ($m^3\,s^{-1}$); groundwater exp(unitless) | Sivapalan et al., 1996 | 403;1.5 |
| Saturated hydraulic conductivity (mm/day) | Calculated based on soil types forest coverage and Price et al., 2010 | 226.6 |
