# Peer review of "Frequency and magnitude variability of Yalu River flooding: Numerical analyses for the last 1000 years"

_Hydrology and Earth System Sciences, 2019_

## Referee Comment (RC1) · Anonymous Referee #1 · 8 Mar 2020

The paper coupled HYDROTREND with the ECHO-G model to reconstruct and investigate the impacts of climate change and human activity on the flooding frequency and magnitude for the Yalu River over the past 1000 years. The results indicated that the frequency trends of flooding were dominated (increased) by climate variability, i.e., intensity and frequency of rainfall events. The also found that deforestation increased the magnitude of floods by 19.2~20.3%, while the construction of cascade reservoirs significantly reduced their magnitude by 36.7~41.7%. In general, the paper presents some useful analyses and can potentially make a useful contribution to the field. However, there are some critical issues need to be addressed. All the major and minor issues I found are included in the detailed review below.

[Figure]

Major comments:

(1) According to section 3.2, the climate model ECHOG was used to simulate monthly precipitation and temperature of Yalu River over last millennium. How to calibrate by meteorological station data? The accuracy of the simulated precipitation and temperature would have an important impact on flood simulation by HYDROTREND model. If there are large biases in ECHOG simulation, a bias correction is necessary before coupled with HYDROTREND model. But there is no relevant information in the paper. On the other hand, the HYDROTREND model was run at the daily scale (as shown in Figures 3d and 4d), whereas the precipitation and temperature of ECHOG are simulated at the monthly scale. How the authors downscaled monthly-scale climate data to daily scale. The authors should provide relevant information in detail.

(2) The authors used the GEV distribution to calculate the return interval flood values. How to estimate the parameters of the GEV when fitting the data of peak flows? There is no any information about it. In addition, I am not sure if the GEV is the best distribution for the study basin, which raises another key question: why not use other distribution functions such as P-3, since the P-3 is widely used for the frequency analysis of floods in Chinese basins. Or, why not use multiple probability distributions and find the optimal distribution to analyze flood frequency? The authors need to carefully clarify this.

(3) The flooding frequency analysis is based on the hydrological model coupled with the climate model. In my opinion, there would be large uncertainties throughout the process of modeling and frequency function analysis as well as the data used, especially for such long-term (1000 years) hydrological simulations. The authors should make a discussion to emphasize this point.

Specific comments:

Line 15: what's the meaning of "AD"? Please give it full name.

Line 21: what' s the meaning of "larger floods"? please clarify it.

Lines 228-230: Please provide information about the spatial resolution of the ECHO-G model.

Lin 250: How to identify wet years, average years and dry years? please clarify it.

Line 255: Why use 14 years as the period of wet and dry years for the Yalu River basin?

Table 1: Which basin's error results are summarized in Table 1? Ai River or Yalu River? please clarify it.

Figure 3: the performance of model seems not well for daily peak flows in the Ai River, how would this affect the flood frequency analysis?

Figures 3 and 4: I suggest the x-axis of (e), (f), (g) in Figures 3 and 4 be marked with the actual year.

Section 4.3: wavelet analysis is conducted based on continuous (flood) data over a certain period. How to compute the long-term (1000-2012) series of the designed floods with different return intervals? As I know, for a specific timeseries there is only one value for a certain return period fitted by the GEV distribution. How to generate a long-term data of the designed floods used for wavelet analysis? please clarify it.

Table 3: how to calculate the frequency of flood occurrence for different recurrence intervals? More explanations are needed.

Line 458: decreased » increased

---

## Referee Comment (RC2) · Anonymous Referee #2 · 12 Mar 2020

The manuscript offers a very interesting and comprehensive study of changes in flood magnitude and frequency over the last 1000 years in Yalu River, China. The manuscript is overall well written but could be shortened by removing some redundancies. Some of the methodologies and results are unclear and must be better explained (see below). I found it quite challenging to assess the robustness of the analysis given the limitations in the model and input data. The use of monthly precipitation with daily simulations based on probability distribution for flood analysis is problematic, and, to a lesser extent, the reliance on peak annual discharge results. The model simplistic anthropogenic representation is also a confining factor. The authors are clearly aware of these limitations and effectively mitigated these limitations by framing the bulk of the

analysis on differences between long time-periods. The authors need to more clearly and explicitly discuss these limitations early in the manuscript.

Specific Comments:

Line 74 - 'become' can be removed

The sentence starting in line 258 - Appendix A2 does not seem to show that.

Section 3.4 - additional information will be helpful - how was it calculated? what values were used (Qpeak)?

Section 4.1.1. - qualitative results are very limited which always raises suspicion unless clearly justified.

Figures 3 and 4 - consider changing the x axis title from 'item' to 'ranked yearly peak flow'. It may be worthwhile trying to explain or at least speculate about years in which the simulated and observed Qpeak strongly diverge.

Line 351 and later - 'frequencies of immense floods of 22.0%...' it is not clear to me what 22% means in this context! It is crucial that the authors clarify this as it is one of the main quantitative metrics used in the manuscript.

Line 407 - 'observed' - I think 'estimated' is more appropriate.

Lines 421-422 - this seems to be a bit too specific an explanation given the model's limited anthropogenic representation. Figure 8 and associated text - the figure needs to be better explained. It is not at all clear what it is showing.

Figure 9 and associated text - the figure is not immediately clear and could benefit from an explanation on how to interpret it. The results drawn from it are not at all apparent e.g. line 473, 479 & 487.

Line 500 - 'flood magnitudes' - Figure 10 title is % frequency of floods - is it magnitude or frequency?

Lines 538-542- is this based on the results or general assertion?

Line 544 and elsewhere - 'coupled' may be misleading in this context because it implies that the two models are dynamically coupled where, to my understanding, the output of ECHO-G is used as input dataset to HYDROTREND (just like any other input dataset).

———————————————

---

## Author Comment (AC1) · 21 Apr 2020

The paper coupled HYDROTREND with the ECHO-G model to reconstruct and investigate the impacts of climate change and human activity on the flooding frequency and magnitude for the Yalu River over the past 1000 years. The results indicated that the frequency trends of flooding were dominated (increased) by climate variability, i.e., intensity and frequency of rainfall events. The also found that deforestation increased the magnitude of floods by 19.2-20.3 percent while the construction of cascade reservoirs significantly reduced their magnitude by 36.7-41.7 percent. In general, the paper presents some useful analyses and can potentially make a useful contribution to the

field. However, there are some critical issues need to be addressed. All the major and minor issues I found are included in the detailed review below.

Dear review: We greatly appreciate for your positive summary on our study, your comments and valuable suggestions are helpful for us to improve the manuscript. The manuscript has been carefully revised and point-by-point responses are listed as below.

Major comments:

Q1: (1) According to section 3.2, the climate model ECHOG was used to simulate monthly precipitation and temperature of Yalu River over last millennium. How to calibrate by meteorological station data? The accuracy of the simulated precipitation and temperature would have an important impact on flood simulation by HYDROTREND model. If there are large biases in ECHOG simulation, a bias correction is necessary before coupled with HYDROTREND model. But there is no relevant information in the paper.

Response: I agree that in order to convince readers for availability of simulated stream flow values, clarifying accuracy of climate data over the last 1000 years is essential. In response to review's comments, Fig.3 has been added and the processes of calibration and bias corrections for climate data in past 1000 years have been clarified based on meteorological station data.

Line 232-246 in revised manuscript: As shown in Fig. 3, ECHO-G can accurately predict the actual variations in temperatures of the Yalu River, and additionally, it can accurately capture the inter-annual seasonal precipitation distribution. However, there was a certain bias in the observed and simulated annual precipitation when comparing the ranked multi-year precipitations, where data were significantly dominated by the simulated precipitation. The calibrated and validated relationship between simulations and bias of precipitation during 1957–1990 was applied to modify the annual simulated precipitation over the last millennium, where amplitudes of simulated precipitation during 1957–1990 basically covered the whole simulated period (Fig. 3). The climate data for the Ai River over the past millennium were also modified through the monthly relationship of the Yalu's and Ai's temperature and precipitation during 1957–2012.

Q2: On the other hand, the HYDROTREND model was run at the daily scale (as shown in Figures 3d and 4d), whereas the precipitation and temperature of ECHOG are simulated at the monthly scaled. How the authors downscaled monthly-scale climate data to daily scale. The authors should provide relevant information in detail.

Response: Thank you for your professional comments. Similar to most of Downscaled Global Climate Model (GCM) forced with a variety of emissions scenarios generates daily resolution outputs based on Monte Carlo analysis, the rainfall event module and degree-day module in HYDROTREND downscaled monthly precipitation and temperature to daily scale through the same methods. In order to help reader effectively understand how monthly-scale climate data downscaled, we first add the summary of solution in first pages and given some details in method descriptions (Line 86-89 and Line 199-210 in revised manuscript).

Q3: (2) The authors used the GEV distribution to calculate the return interval flood values. How to estimate the parameters of the GEV when fitting the data of peak flows? There is no any information about it. In addition, I am not sure if the GEV is the best distribution for the study basin, which raises another key question: why not use other distribution functions such as P-3, since the P-3 is widely used for the frequency analysis of floods in Chinese basins. Or, why not use multiple probability distributions and find the optimal distribution to analyze flood frequency? The authors need to carefully clarify this.

Response: We have followed your suggestion. In the new manuscript, the reasons for why use GEV distribution combined with block maxima method were clarified. For frequency analysis of floods in Yalu River, the GEV distribution combined with block maxima method and P-III distribution (widely used in Chinese basins) were compared

to study the impact of the two methods on research targets of the paper, parameters of distributions were estimated by L-moments method. In addition, we also gave some descriptions for the block maxima method, which was applied in this paper to reduce the uncertainties of simulations. Response to the suggestion Appendix A4 has been added and this section has been revised as follow :

Line 304-317 in revised manuscript:The generalized extreme-value distribution (GEV) and Pearson type three distribution (P-III) combined with the L-moments method have been widely used to investigate flood characteristics, of which P-III has been widely adopted for the frequency analysis of floods in many Chinese rivers (Xu et al., 2016). For this study region, GEV based on the block maxima method and P-III showed significant differences for flood estimations on return periods larger than the observed time periods (1958–2012 for 55 years)( Appendix A4.a-b). However, two methods have a little difference for investigating the impact of climate change and human activities on 100-, 50-, 20- and 10-year floods when samples increased to 1000 years generated by model (Appendix A4.c). In addition, the block maxima method in GEV, which divides the estimations period into non-overlapping periods of equal size and restricts attention to the maximum estimations in each period, can reduce the uncertainties of simulations (Ferreira and Laurens, 2015). Therefore, here the L-moments method for parameter estimation of the GEV was applied to study the flood frequency in the Yalu River based on simulated annual peak discharges in Yalu River, combined with the block maxima method.

Q4: (3) The flooding frequency analysis is based on the hydrological model coupled with the climate model. In my opinion, there would be large uncertainties throughout the process of modeling and frequency function analysis as well as the data used, especially for such long-term (1000 years) hydrological simulations. The authors should make a discussion to emphasize this point.

Response: HYDROTREND have limitations for simulating annual peak flows over the last 1000 years due to the uncertainties of input boundary conditions and model assumptions. In the revised manuscript, we firstly discussed limitations induced by model assumptions, uncertainties of climate data input and limitations caused by simplistic anthropogenic impacts, respectively, and then some descriptions are given to clarify how to reduce the uncertainty of simulation results in this paper.

Specific comments:

Q5:Line 15: what's the meaning of "AD"? Please give it full name.

Response: A.D. is a Latin abbreviation for Anno Domini, in the year of our Lord. A.D. is used with dates in the current era, corresponding to B.C (before Christ indicates that a date is before the Christian era). When there is no A.D or B.C mark before the era, the time means current era. In the revised manuscript, all A.D marks before the year were removed.

Q6:Line 21: what's the meaning of "larger floods"? please clarify it.

Response: We have added specific designed floods (100- and 50-year) for "larger floods".

Q7: Lines 228-230: Please provide information about the spatial resolution of the ECHO-G model.

Response: We have been followed this suggestion in revised manuscript.

Q8: Line 250: How to identify wet years, average years and dry years? please clarify it.

Response: In this paper, different rainfall periods (wet, average and dry years) in Yalu and Ai rivers were defined based on observed climate data during 1958-2012. Classified total rainfall patterns were applied to calibrate rainfall event distribution coefficients and exponents which are significant model input parameters strongly correlated with the simulated daily rainfall events. The results of calibration were used to reconstruct the annual maximum water discharge over the last 1000 years, combining with

long-term input boundaries. The Appendix A3 was added for in response to review's suggestion.

Q9: Line 255: Why use 14 years as the period of wet and dry years for the Yalu River basin?

Response: The period of wet and dry years for the Yalu River basin obtained from Yi et al., 2014. The results of this paper indicated that Yalu River and its adjacent rivers (Liaohe River, small and medium-sized rivers along the east coast of Liaodong peninsula, Songhua River, etc.) have periodic of 14 year for wet years and dry years based on analysis of multi-years monitoring hydrological data. Periodic of 14 years as the time unit of simulation can effectively improve the accuracy of daily rainfall events simulation, combing with different rainfall conditions. In the process of model simulation, the estimated precipitation over the last 1000 years was first divided into multiple consecutive 14 years, and then model input parameters, strongly correlated with the simulated daily rainfall events, were adjusted according to the classification criteria of rainfall patterns and rainfall data for 14 years. This process can reduce the uncertainties of simulations induced by climate data. Multiple input files of modeling are generated by R Programming Language, and multiple simulation process were conducted through script editing.

Q10: Table 1: Which basin's error results are summarized in Table 1? Ai River or Yalu River? please clarify it.

Response: This question is reply together with the next one. The summary of error for design floods in Table1 is from Yalu River in original manuscript. However, in the revised manuscript Table1 was replaced by figure of Appendix A4. As shown in Appendix A4.a-b, flood frequency analysis was conducted in Yalu and Ai rivers based on simulated and observed peak discharges during 1958-2012, combined with GEV and P-III distribution. The results show that the model can simulate the changes of flood frequencies in Yalu and Ai rivers. Although the simulation results of Ai River are slightly

inferior to those of Yalu River, it has no significant difference for investigating the impact of climate change and human activities on flood frequencies (100-, 50-, 20-year, etc.).

Q11: Figure 3: the performance of model seems not well for daily peak flows in the Ai River, how would this affect the flood frequency analysis?

Response: Please refer to above reply for this question.

Q12: Figures 3 and 4: I suggest the x-axis of (e), (f), (g) in Figures 3 and 4 be marked with the actual year.

Response: We have followed your suggestion to make changes for figures.

Q13: Section 4.3: wavelet analysis is conducted based on continuous (flood) data over a certain period. How to compute the long-term (1000-2012) series of the designed floods with different return intervals? As I know, for a specific time series there is only one value for a certain return period fitted by the GEV distribution. How to generate a long-term data of the designed floods used for wavelet analysis? please clarify it.

Response: Thank you for your suggestions. This section was used to indicate the qualitative impact of climate change and human activities on flood frequency. We first set standards based on design floods estimated by simulated annual peak discharges during 1000-2012. And then, thresholding process was conducted to produces new data sets (over standards for 1, otherwise for 0), based on times series of peak discharges over the last 1000 years and standards. Next, wavelet analysis was conducted for new data sets to produce times series of the occurrence frequencies of floods exceeding different return period standards. Eventually, the results were applied to qualitative analysis the impact of climate change and human activities on flood frequency. The descriptions for this section are insufficient in original manuscript, which led to the confusion. This section has been rewritten in response to review's suggestions (Line 492-525 in revised manuscript).

Q14: Table 3: how to calculate the frequency of flood occurrence for different recurrence intervals? More explanations are needed.

Response: In the original manuscript, the ratio of the number of simulated flood peaks exceeding the standards to the statistical years is defined as 'frequency of flood occurrence', and designed floods of different recurrence intervals estimated by simulated annual peak discharges during 1000-2012 were applied for the standards. In the revised manuscript, Table3 and the Sections related to 'frequency of flood occurrence' were removed. The reasons are as follows: 1): Distinction between the frequency of floods and 'frequency of flood occurrence' cause confusions of readers. 2) The impacts of climate change and human activities on floods are able to clarify through discussing the changes of frequencies of floods or magnitudes of design floods. 3) The manuscript is more shortened and clear by removing related redundancies.

Q15: Line 485: decreased corrected to increased

Response: Line 553: Revised.

––––––––––––––––––––––––––––––

[Figure]

**Fig. 1.** Figure 3. Correction of the simulated climate data from the ECHO-G model based on observations during 1957–1990: (a) annual ranked precipitation distribution of observations, simulations, and the gap;

[Figure]

Buffer zone = statistic years / 3

lowest rainfall          location of average          peak rainfall

dry years          average years          wet years

Buffer zone

Ranked yearly precipatation during 1958-2012 (55 years)

**Fig. 2.** Appendix A3.The classification method for different rainfall conditions (wet, average and dry years) in Yalu and Ai rivers

[Figure]

[Figure]

**Fig. 3.** Appendix A4. Comparison between the observed and simulated return interval peak discharges in the Ai River and Yalu River based on the GEV and P-III methods. The design floods for the period 1958–2012

---

## Author Comment (AC2) · 21 Apr 2020

The manuscript offers a very interesting and comprehensive study of changes in flood magnitude and frequency over the last 1000 years in Yalu River, China.

Dear review: We greatly appreciate for your positive comments on our study. Your valuable suggestions are helpful for us to improve the manuscript. The manuscript has been carefully revised and point-by-point responses are listed as below.

Q1: The manuscript is overall well written but could be shortened by removing some redundancies. Some of the methodologies and results are unclear and must be better

[Figure]

explained (see below).

Response: We appreciate for this important suggestion. Response to review's suggestion the methodologies are modified as follows:

1) The model description was simplified, and some details for rainfall event module and degree-day module in HYDROTREND were given which can help reader effectively understand how monthly-scale climate data downscaled to daily –scale (Line171-228 in revised manuscript).

2) The accuracy of the simulated precipitation and temperature from ECHOG outputs were clarified (Line232-246 and Line 25 in revised manuscript).

3) Why use the GEV distribution to calculate the return interval flood values, and How to estimate the parameters of the GEV were clarified (Line 304-317 and Line772 in revised manuscript).

Response to review's suggestion the results are modified as follows:

1): Model limitations induced by model assumptions, uncertainties of climate data input and simplistic anthropogenic impacts were clarified (Line 406-422 in revised manuscript).

2): The results for investigating factors controlling the flood frequency variability have been shortened. Meanwhile, the Section for qualitative study impact of climate change and human activities on flood frequency has been rewritten (Line 491-571 in revised manuscript).

Q2: I found it quite challenging to assess the robustness of the analysis given the limitations in the model and input data. The use of monthly precipitation with daily simulations based on probability distribution for flood analysis is problematic, and, to a lesser extent, the reliance on peak annual discharge results.

Response: Thank you for your professional comments. Different from the precise flood

forecasting model system with high temporal and spatial resolution input boundary conditions (DEMs, climate data, anthropogenic activities, etc.) the model has limitations for simulating annual peak flows over the last 1000 years due to the uncertainties of input boundary conditions and model assumptions. In order to reduce the uncertainty of simulation results, monthly scale climate data from ECHO-G outputs were downscaled to daily scale based on rainfall event module and degree-day module in the model combining with Monte-Carlo technique. Meanwhile, different multi-rainfall patterns (total of nine categories: wet year-SMW, average year-SMW, and dry year-SMW) were applied to better simulate daily precipitation intensity and distribution in process of modeling. The GEV combined with the block maxima method was adopted to reduce the uncertainty of simulations through improving the quality of reconstructed samples. Furthermore, the bulk of the analysis for flood characteristics in special periods with different climate and human activities was conducted to mitigate the impacts of simplified boundary conditions.

This paper provides an attempt to: 1. improve the accuracy of design floods by expanding the samples of historical floods. 2. Investigate the impacts of climate change and human activities (deforestation and dams) on floods by comparing flood characteristics in different periods which have more significant differences in climate characteristics and human activities. How to further improve the limitations of the model (for example, input of spatially changing precipitations, higher resolution for climate data, DEMs and more complex of anthropogenic activities) is worth study in the future.

Q3: The model simplistic anthropogenic representation is also a confining factor. The authors are clearly aware of these limitations and effectively mitigated these limitations by framing the bulk of the analysis on differences between long time-periods. The authors need to more clearly and explicitly discuss these limitations early in the manuscript.

Response: We have followed your suggestion. In revised manuscript, the descriptions for model limitations and uncertainties, and how to reduce the uncertainties of simulations were added (Line 406-422 in revised manuscript).

Specific Comments:

Q4: Line 74 - 'become' can be removed

Response: Revised

Q5: The sentence starting in line 258 - Appendix A2 does not seem to show that.

Response: We have followed your suggestion. The related reference was added.

Q6: Section 3.4 - additional information will be helpful - how was it calculated? What values were used (Qpeak)?

Response: In revised manuscript, Appendix A4 and related descriptions have been added to clarify why use the GEV distribution to calculate the return interval flood values, and how to estimate the parameters of the GEV In this paper, the L-moments method for parameter estimation of the GEV was applied to study the flood frequency in the Yalu River based on simulated annual peak discharges in Yalu River, combined with the block maxima method (Line 304-317 in revised manuscript).

Q7: Section 4.1.1. - qualitative results are very limited which always raises suspicion unless clearly justified.

Response: The descriptions for this section are insufficient in original manuscript, which led to the confusion. This section has been rewritten in response to review's suggestions. Response to this comment, this section has been revised as follow:

4.4.1 Qualitative flooding frequency analysis in response to basin changes

Line 492-525: Simulated annual peak discharge including impacts of climate change and human activities were thresholding processed (over threshold for 1 and otherwise 0) based on design floods level of different flood return intervals over the past 1000 years, and the same process was adopted for annual rainfall based on the standard of extreme rainfall events (strong rainfall in wet years greater 942 mm yr-1) in the Yalu River, as shown in Appendix A1. Generated time series datasets were conducted by using a wavelet analysis to qualitatively investigate the dominant controls on flood frequency variability for the Yalu River over the past 1000 years (Fig. 9). The wavelet results showed that during 1130–1190, 1280–1340, 1520–1580, and 1880–1940, the occurrence frequencies of floods exceeding the 50-years return period standard were much higher than those of other periods, and related extreme rainfall events also showed similar trends (Fig. 9). The occurrence frequency of floods over the 50–year standard during 1000–1450 was close to LIA (1450–1840), similar to the intensity and frequency of extreme rainfall events. In contrast, occurrence frequencies of floods over the 20–year and 10-year standards during 1000–1450 were much higher than that of the LIA, which was more related to the variations of multi-year average precipitation (Fig. 9). Compare with LIA, occurrence frequencies of floods over 50 years during 1841–1940 rapidly increased, and occurrence frequencies of floods over 10–years was basically at the same level in response to the significant increasing intensity and frequency of extreme rainfall events and similar average annual rainfall (Fig. 9). Our results demonstrate that the frequency and intensity of extreme precipitation caused by climate change have a dominant control on the frequencies of large floods (100-year, 50-year). However, medium and small-magnitude floods (20-year, 10-year, and 5-year) are more closely linked to long-term climatic trends of warming and humidity (Figs. 2 and 9).

As shown by Fig. 9, the occurrence frequencies of floods over different return interval standards rapidly decreased after 1940 due to the construction of cascading reservoirs, despite the increasing frequency and intensity of extreme precipitation events in response to climate change and anthropogenic impacts. The results demonstrate that the construction of reservoirs can effectively reduce flood disasters for the Yalu River basin despite having little effect on the long-term runoff to the sea (Sheng et al., 2019); additionally, the declines of occurrence frequencies for medium- and small-magnitude floods (20 year, 10 year) predominated over those of large floods (50 year) due to the

construction of flood retention dams.

Q8: Figures 3 and 4 - consider changing the x axis title from 'item' to 'ranked yearly peak flow'. It may be worthwhile trying to explain or at least speculate about years in which the simulated and observed Qpeak strongly diverge.

Response: We have followed your suggestion to make changes for figures. The descriptions for simulated and observed Qpeak strongly diverge and its impacts for investigating frequencies of floods were given in revised manuscript. The Table1(in original manuscript) for comparison between observed and simulated return interval peak discharges for the period 1958–2012 in Yalu River were replaced by Appendix A4 in new manuscript. The picture can not only clarify why GEV was used to estimate design floods, but also explain the accuracy of the simulation results for flood frequency analysis.

Q9: Line 351 and later - 'frequencies of immense floods of 22.0%...' it is not clear to me what 22% means in this context! It is crucial that the authors clarify this as it is one of the main quantitative metrics used in the manuscript.

Response: The terms of 'Frequencies of immense floods of 22.0%' means the numbers of recorded immense floods per 100 years was 22. Inaccuracy of descriptions caused confusion of readers, and this section and related Table have been revised in the new manuscript.

Q10: Line 407 - 'observed' - I think 'estimated' is more appropriate.

Response: Revised.

Q11: Lines 421-422 - this seems to be a bit too specific an explanation given the model's limited anthropogenic representation.

Response: We have followed your suggestion.

Q12: Figure 8 and associated text - the figure needs to be better explained. It is not at

all clear what it is showing.

Response: This section and caption of this figure have been revised in new manuscript.

Q13: Figure 9 and associated text - the figure is not immediately clear and could benefit from an explanation on how to interpret it. The results drawn from it are not at all apparent e.g. line 473, 479 & 487.

Response: The figure and captions have been revised. In order to clearly show the impact of climate change and human activities (deforestation and dams) on magnitude of design floods, new table has been added.

Q14: Line 500 - 'flood magnitudes' - Figure 10 title is % frequency of floods - is it magnitude or frequency?

Response: In the original manuscript, the ratio of the number of simulated flood peaks exceeding the standards to the statistical years is defined as 'frequency of flood occurrence', and designed floods of different recurrence intervals estimated by simulated annual peak discharges during 1000-2012 were applied for the standards. Distinguishing frequency of floods and 'frequency of flood occurrence' easily make reader confusions. Actually, the impacts of climate change and human activities on floods are able to clarify through discussing the changes of frequencies of floods or magnitudes of design floods. Therefore, we removed the Sections related to 'frequency of flood occurrence' to dispel confusions and redundancies.

Q15: Lines 538-542- is this based on the results or general assertion?

Response: It is general assertion based on previous researches (a change in global or regional climate patterns; hydrological characteristics of medium and small rivers) and conclusions of this paper. We would like to let the readers to know possible changing trend of floods in the future through this assertion. We are not to make conclusion without careful investigation and available data.

Q16: Line 544 and elsewhere - 'coupled' may be misleading in this context because it

implies that the two models are dynamically coupled where, to my understanding, the output of ECHO-G is used as input dataset to HYDROTREND (just like any other input dataset).

Response: We have followed this suggestion in the revised manuscript. We agree that 'coupled' seems like to imply the interactions between two model systems, similar to a coupled model of waves and tides in the ocean. In this study output of ECHO-G is only used to as input dataset.

[Figure]

**Fig. 1.** Appendix A4. Comparison between the observed and simulated return interval peak discharges in the Ai River and Yalu River based on the GEV and P-III methods. The design floods for the period 1958–2012

---

## Author Response (AR1)

**Response to Editor**

The referees provided a number of constructive remarks on the points that should be addressed, since important clarifications are needed.

Dear editor: The manuscript has been carefully revised and point-by-point responses for professional suggestions of referees are conducted. Point-by-point responses for editor's comments are also listed as below in order to more effective review for this revised manuscript.

Q: As I highlighted also in my first comments, the meteorological forcing is crucial and the main novelty in this work, so I fully agree with Ref#1 (Comment 1) that more detail is needed, in particular on the bias correction method given the coarse spatial scale of the ECHO-G model and on the downscaling procedure,

Response:

**Line 228-244 and Line 260-265 in revised manuscript:** We have followed this significant suggestion. In revised manuscript, the accuracy and bias corrections for climate data in past 1000 years have been clarified based on meteorological station data (Line 281-295 and Line 305-311 in Mark-up manuscript version).

**Line 82-88 and Line 196-208 in revised manuscript:** In order to help reader effectively understand how monthly-scale climate data downscaled, we added the summary of solution in first pages and given some details in method descriptions (Line 83-89 and Line 238-250 in Mark-up manuscript version).

Q: if the GCM model scenarios are at monthly scale (passing from monthly, to daily, to peak values is indeed a challenging task, implying very high uncertainties as stressed by Ref#2, and both referees ask for more discussion on such uncertainty and of the limitations of the study).

Response:

**Line 400-416 in revised manuscript:** In response to this professional comment, Section 4.2 was added to discuss the model limitations and uncertainties. In this section, we first discussed model limitations induced by model assumptions, uncertainties of climate data input and limitations caused by simplistic anthropogenic impacts, respectively, and then some descriptions are given to clarify how to reduce the uncertainties of simulation results in this paper. Under multi-rainfall patterns, the daily rainfall events generated by the Monte-Carlo technique combined with monthly climate input data (ECHO-G model output) were applied in this model to reduce the uncertainties of climate data (This process is equivalent to refining the input climate data). Meanwhile, the bulk of the analysis for flood characteristics in special periods with different climate and human activities was conducted to mitigate the impacts of simplified boundary conditions (more significant changes for climatic characteristics and human activities in past 1000 years) (Line 471-487 in Mark-up manuscript version).

Q: And also more info on the agreement between the downscaled and bias-corrected grids with the available ground measures at least for the overlapping observation period. I invite you to submit a revised version, that I will ask to at least one referee to review again.

Response:

**Line 338-354 revised manuscript:** We have followed this suggestion. Under multi-rainfall patterns, daily rainfall events generated by the Monte-Carlo technique combined with monthly climate input data were applied to driven this model to simulate the annual daily peak flow in Yalu and Ai River during 1958-2012 (passing from monthly, to daily, to peak values). Compare with observed data during 1958-2012, we discussed the performance of simulations including the limitations, uncertainties, and the impacts of research objectives (Line 395-416 in Mark-up manuscript version).

**Response to review1**

The paper coupled HYDROTREND with the ECHO-G model to reconstruct and investigate the impacts of climate change and human activity on the flooding frequency and magnitude for the Yalu River over the past 1000 years. The results indicated that the frequency trends of flooding were dominated (increased) by climate variability, i.e., intensity and frequency of rainfall events. The also found that deforestation increased the magnitude of floods by 19.2~20.3%, while the construction of cascade reservoirs significantly reduced their magnitude by 36.7~41.7%. In general, the paper presents some useful analyses and can potentially make a useful contribution to the field. However, there are some critical issues need to be addressed. All the major and minor issues I found are included in the detailed review below.

Dear review: We greatly appreciate for your positive summary on our study, your comments and valuable suggestions are helpful for us to improve the manuscript. The manuscript has been carefully revised and point-by-point responses are listed as below.

Major comments:

Q: (1) According to section 3.2, the climate model ECHOG was used to simulate monthly precipitation and temperature of Yalu River over last millennium. How to calibrate by meteorological station data? The accuracy of the simulated precipitation and temperature would have an important impact on flood simulation by HYDROTREND model. If there are large biases in ECHOG simulation, a bias correction is necessary before coupled with HYDROTREND model. But there is no relevant information in the paper.

Response:

**Line 228-244 and Line 260-265 in revised manuscript:** I agree that in order to convince readers for availability of simulated stream flow values, clarifying accuracy of climate data over the last 1000 years is essential. In response to review's comments, the processes of calibration and bias corrections for climate data in past 1000 years have been clarified based on meteorological station data (Line 281-295 and Line 305-311 in Mark-up manuscript version).

[Figure]

Figure 3. Correction of the simulated climate data from the ECHO-G model based on observations during 1957–1990: (a) annual ranked precipitation distribution of observations, simulations, and the gap; (b) the relationship between simulations and the gap for the period of 1957–1970 and 1977–1990; (c) calibration (1957–1970 and 1977–1990) and validation results (1967–1980); (d) monthly measured and simulated rainfall percentage; (e) and (f) comparison of the simulated and observed temperatures during 1957–1990.

Q: On the other hand, the HYDROTREND model was run at the daily scale (as shown in Figures 3d and 4d), whereas the precipitation and temperature of ECHOG are simulated at the monthly scaled. How the authors downscaled monthly-scale climate data to daily scale. The authors should provide relevant information in detail.

Response:

**Line 82-88 and Line 196-208 in revised manuscript:** Thank you for your professional comments. Similar to most of Downscaled Global Climate Model (GCM) forced with a variety of emissions scenarios generates daily resolution outputs based on Monte Carlo analysis, the rainfall event module

and degree-day module in HYDROTREND downscaled monthly precipitation and temperature to daily scale through the same methods. In order to help reader effectively understand how monthly-scale climate data downscaled, we first add the summary of solution in first pages and given some details in method descriptions (Line 83-89 and Line 238-250 in Mark-up manuscript version).

Q: (2) The authors used the GEV distribution to calculate the return interval flood values. How to estimate the parameters of the GEV when fitting the data of peak flows? There is no any information about it. In addition, I am not sure if the GEV is the best distribution for the study basin, which raises another key question: why not use other distribution functions such as P-3, since the P-3 is widely used for the frequency analysis of floods in Chinese basins. Or, why not use multiple probability distributions and find the optimal distribution to analyze flood frequency? The authors need to carefully clarify this.
Response:
**Line 301-314 and Line 764-767 in revised manuscript:** We have followed your suggestion. In the new manuscript, the reasons for why use GEV distribution combined with block maxima method were clarified. For frequency analysis of floods in Yalu River, the GEV distribution combined with block maxima method and P-III distribution (widely used in Chinese basins) were compared to study the impact of the two methods on research targets of the paper, parameters of distributions were estimated by L-moments method. In addition, we also gave some descriptions for the block maxima method, which was applied in this paper to reduce the uncertainties of simulations (Line 355-368 and Line 890-893 in Mark-up manuscript version).

[Figure]

A4. Comparison between the observed and simulated return interval peak discharges in the Ai River and Yalu River based on the GEV and P-III methods. The design floods for the period 1958–2012 in Ai River (a) and Yalu River (b), and (c) The design floods for the period 1000-2012 in total Yalu River.

Q: (3) The flooding frequency analysis is based on the hydrological model coupled with the climate model. In my opinion, there would be large uncertainties throughout the process of modeling and frequency function analysis as well as the data used, especially for such long-term (1000 years) hydrological simulations. The authors should make a discussion to emphasize this point.

Response:

**Line 400-416 in revised manuscript:** HYDROTREND have limitations for simulating annual peak flows over the last 1000 years due to the uncertainties of input boundary conditions and model assumptions. In the revised manuscript, we firstly discussed limitations induced by model assumptions, uncertainties of climate data input and limitations caused by simplistic anthropogenic impacts, respectively, and then some descriptions are given to clarify how to reduce the uncertainty of simulation results in this paper (Line 471-487 in Mark-up manuscript version).

Specific comments:

Q:Line 15: what's the meaning of "AD"? Please give it full name.

Response:

A.D. is a Latin abbreviation for Anno Domini, in the year of our Lord. A.D. is used with dates in the current era, corresponding to B.C (before Christ indicates that a date is before the Christian era). When there is no A.D or B.C mark before the era, the time means current era. In the revised manuscript, all A.D marks before the year were removed.

Q: Line 21: what's the meaning of "larger floods"? please clarify it.

Response:

**Line 22 in revised manuscript:** We have added specific designed floods for "larger floods" (Line 22 in Mark-up manuscript version).

Q: Lines 228-230: Please provide information about the spatial resolution of the ECHO-G model.

Response:

**Line 231-233 in revised manuscript:** We have been followed this suggestion in revised manuscript (Line 281-283 in Mark-up manuscript version).

Q: Line 250: How to identify wet years, average years and dry years? please clarify it.

Response:

**Line 759-761 in revised manuscript:** In this paper, different rainfall periods (wet, average and dry years) in Yalu and Ai rivers were defined based on observed climate data during 1958-2012. Classified total rainfall patterns were applied to calibrate rainfall event distribution coefficients and exponents which are significant model input parameters strongly correlated with the simulated daily rainfall events. The results of calibration were used to reconstruct the annual maximum water discharge over the last 1000 years, combining with long-term input boundaries. The Appendix A3 was added for in response to review's suggestion (Line 885-887 in Mark-up manuscript version).

[Figure]

Buffer zone = statistic years / 3

lowest rainfall          location of average          peak rainfall

Ranked yearly precipatation during 1958-2012 (55 years)

A3.The classification method for different rainfall conditions (wet, average and dry years) in Yalu and Ai rivers

Q: Line 255: Why use 14 years as the period of wet and dry years for the Yalu River basin?

Response:

**Line 278 in revised manuscript:** The period of wet and dry years for the Yalu River basin obtained from Yi et al., 2014. The results of this paper indicated that Yalu River and its adjacent rivers (Liaohe River, small and medium-sized rivers along the east coast of Liaodong peninsula, Songhua River, etc.) have periodic of 14 year for wet years and dry years based on analysis of multi-years monitoring hydrological data. Periodic of 14 years as the time unit of simulation can effectively improve the accuracy of daily rainfall events simulation, combing with different rainfall conditions. In the process of model simulation, the estimated precipitation over the last 1000 years was first divided into multiple consecutive 14 years, and then model input parameters, strongly correlated with the simulated daily rainfall events, were adjusted according to the classification criteria of rainfall patterns and rainfall data for 14 years. This process can reduce the uncertainties of simulations induced by climate data. Multiple input files of modeling are generated by R Programming Language, and multiple simulation process were conducted through script editing (Line 333 in Mark-up manuscript version).

Q: Table 1: Which basin's error results are summarized in Table 1? Ai River or Yalu River? please clarify it.

Response:

**Line 343-354 and Line 764-767 in revised manuscript:** This question is reply together with the next one. The summary of error for design floods in Table1 is from Yalu River in original manuscript. However, in the revised manuscript Table1 was replaced by figure of Appendix A4. As shown in Appendix A4.a-b, flood frequency analysis was conducted in Yalu and Ai rivers based on simulated and observed peak discharges during 1958-2012, combined with GEV and P-III distribution. The results

show that the model can simulate the changes of flood frequencies in Yalu and Ai rivers. Although the simulation results of Ai River are slightly inferior to those of Yalu River, it has no significant difference for investigating the impact of climate change and human activities on flood frequencies (100-, 50-, 20-year, etc.)(Line 400-414 and Line 890-893 in Mark-up manuscript version).

[Figure]

A4. Comparison between the observed and simulated return interval peak discharges in the Ai River and Yalu River based on the GEV and P-III methods. The design floods for the period 1958–2012 in Ai River (a) and Yalu River (b), and (c) The design floods for the period 1000-2012 in total Yalu River.

Q: Figure 3: the performance of model seems not well for daily peak flows in the Ai River, how would this affect the flood frequency analysis?
Response:
**Line 343-354 and Line 764-767 in revised manuscript:** Figure 3 in original manuscript has been changed to Figure 4 in revised manuscript, because new figure was added. Please refer to above reply for this question (Line 400-414 and Line 890-893 in Mark-up manuscript version).

Q: Figures 3 and 4: I suggest the x-axis of (e), (f), (g) in Figures 3 and 4 be marked with the actual year.

Response:

**Line 355 and Line 361 in revised manuscript:** Figures 3 and 4 in original manuscript have been changed to Figures 4 and 5 in revised manuscript. We have followed your suggestion to make changes for figures (Line 418 and Line 424 in Mark-up manuscript version).

Q: Section 4.3: wavelet analysis is conducted based on continuous (flood) data over a certain period. How to compute the long-term (1000-2012) series of the designed floods with different return intervals? As I know, for a specific time series there is only one value for a certain return period fitted by the GEV distribution. How to generate a long-term data of the designed floods used for wavelet analysis? please clarify it.

Response:

**Line 484-517 in revised manuscript:** Thank you for your suggestions. This section was used to indicate the qualitative impact of climate change and human activities on flood frequency. We first set standards based on design floods estimated by simulated annual peak discharges during 1000-2012. And then, thresholding process was conducted to produces new data sets (over standards for 1, otherwise for 0), based on times series of peak discharges over the last 1000 years and standards. Next, wavelet analysis was conducted for new data sets to produce times series of the occurrence frequencies of floods exceeding different return period standards. Eventually, the results were applied to qualitative analysis the impact of climate change and human activities on flood frequency. The descriptions for this section are insufficient in original manuscript, which led to the confusion. This section has been rewritten in response to review's suggestions (Line 557-607 in Mark-up manuscript version).

Q: Table 3: how to calculate the frequency of flood occurrence for different recurrence intervals? More explanations are needed.

Response:

In the original manuscript, the ratio of the number of simulated flood peaks exceeding the standards to the statistical years is defined as 'frequency of flood occurrence', and designed floods of different recurrence intervals estimated by simulated annual peak discharges during 1000-2012 were applied for the standards. In the revised manuscript, Table3 and the Sections related to 'frequency of flood occurrence' were removed. The reasons are as follows: 1): Distinction between the frequency of floods and 'frequency of flood occurrence' cause confusions of readers. 2) The impacts of climate change and human activities on floods are able to clarify through discussing the changes of frequencies of floods or magnitudes of design floods. 3) The manuscript is more shortened and clear by removing related redundancies (Line 610 and Line 661-684 in Mark-up manuscript version).

Q: Line 485: decreased »increased

Response: Revised (Line 639 in Mark-up manuscript version).

**Response to review2**

The manuscript offers a very interesting and comprehensive study of changes in flood magnitude and frequency over the last 1000 years in Yalu River, China.

Dear review: We greatly appreciate for your positive comments on our study. Your valuable suggestions are helpful for us to improve the manuscript. The manuscript has been carefully revised and Mark-up manuscript version and point-by-point responses are listed as below.

Q: The manuscript is overall well written but could be shortened by removing some redundancies. Some of the methodologies and results are unclear and must be better explained (see below).

Response:

We appreciate for this important suggestion. Response to review's suggestion the methodologies are modified as follows:

1) **Line 168-224 in revised manuscript:** The model description was simplified, and some details for rainfall event module and degree-day module in HYDROTREND were given which can help reader effectively understand how monthly-scale climate data downscaled to daily –scale (Line 180-275 in Mark-up manuscript version).

2) **Line 228-244 in revised manuscript:** The accuracy and bias correction of the simulated precipitation and temperature from ECHOG outputs were clarified (Line 281-295 in Mark-up manuscript version).

3) **Line 301-314 in revised manuscript:** Why use the GEV distribution to calculate the return interval flood values, and How to estimate the parameters of the GEV were clarified (Line 355-368 in Mark-up manuscript version).

Response to review's suggestion the results are modified as follows:

1): **Line 400-416 in revised manuscript:** Model limitations induced by model assumptions, uncertainties of climate data input and simplistic anthropogenic impacts were clarified (Line 471-487 in Mark-up manuscript version).

2): **Line 483-565 in revised manuscript:** The results for investigating factors controlling the flood frequency variability have been shortened. Meanwhile, the Section for qualitative study impact of climate change and human activities on flood frequency has been rewritten (Line 556-685 in Mark-up manuscript version).

Q: I found it quite challenging to assess the robustness of the analysis given the limitations in the model and input data. The use of monthly precipitation with daily simulations based on probability distribution for flood analysis is problematic, and, to a lesser extent, the reliance on peak annual discharge results.

Response:

Thank you for your professional comments. Different from the precise flood forecasting model system with high temporal and spatial resolution input boundary conditions (DEMs, climate data, anthropogenic activities, etc.) the model has limitations for simulating annual peak flows over the last 1000 years due to the uncertainties of input boundary conditions and model assumptions. However, in order to reduce the uncertainty of simulation results, monthly scale climate data from ECHO-G outputs were downscaled to daily scale based on rainfall event module and degree-day module in the model combining with Monte-Carlo technique. Meanwhile, different multi-rainfall patterns (total of nine categories: wet year-SMW, average year-SMW, and dry year-SMW) were applied to better simulate daily precipitation intensity and distribution in process of modeling. The GEV combined with the block maxima method was adopted to reduce the uncertainty of simulations through improving the quality of reconstructed samples. Furthermore, the bulk of the analysis for flood characteristics in special periods with different climate and human activities was conducted to mitigate the impacts of simplified boundary conditions.

This paper provides an attempt to: 1. improve the accuracy of design floods by expanding the samples of historical floods. 2. Investigate the impacts of climate change and human activities (deforestation and dams) on floods by comparing flood characteristics in different periods which have more significant differences in climate characteristics and human activities. How to further improve the limitations of the model (for example, input of spatially changing precipitations, higher resolution for climate data, DEMs and more complex of anthropogenic activities) is worth study in the future.

Q: The model simplistic anthropogenic representation is also a confining factor. The authors are clearly aware of these limitations and effectively mitigated these limitations by framing the bulk of the analysis on differences between long time-periods. The authors need to more clearly and explicitly discuss these limitations early in the manuscript.
Response:
**Line 400-416 in revised manuscript:** We have followed your suggestion. In revised manuscript, the descriptions for model limitations and uncertainties, and how to reduce the uncertainties of simulations were added (Line 471-487 in Mark-up manuscript version).

Q: Specific Comments:
Line 74 - 'become' can be removed
Response: Revised (Line 76 in Mark-up manuscript version).

Q: The sentence starting in line 258 - Appendix A2 does not seem to show that.
Response:
We have followed your suggestion. The related reference was added.

The model input parameters of rainfall event distribution coefficients and exponents were strongly correlated with the simulated daily rainfall (Syvitski et al., 1998) (Line 337 in Mark-up manuscript version).

Q: Section 3.4 - additional information will be helpful - how was it calculated? What values were used (Qpeak)?

Response:

**Line 301-314 in revised manuscript:** In revised manuscript, why use the GEV distribution to calculate the return interval flood values, and How to estimate the parameters of the GEV were clarified. In this paper, the L-moments method for parameter estimation of the GEV was applied to study the flood frequency in the Yalu River based on simulated annual peak discharges in Yalu River, combined with the block maxima method (Line 355-368 in Mark-up manuscript version).

Q: Section 4.1.1. - qualitative results are very limited which always raises suspicion unless clearly justified.

Response:

**Line 484-517 in revised manuscript:** The descriptions for this section are insufficient in original manuscript, which led to the confusion. This section has been rewritten in response to review's suggestions (Line 557-608 in Mark-up manuscript version).

Q: Figures 3 and 4 - consider changing the x axis title from 'item' to 'ranked yearly peak flow'. It may be worthwhile trying to explain or at least speculate about years in which the simulated and observed Qpeak strongly diverge.

Response:

**Line 355 and Line 361 in revised manuscript:** Figures 3 and 4 in original manuscript have been changed to Figures 4 and 5 in revised manuscript. We have followed your suggestion to make changes for figures (Line 417 and Line 424 in Mark-up manuscript version).

**Line 342-354 and Line 767 in revised manuscript:** The descriptions for simulated and observed Qpeak strongly diverge and its impacts for investigating frequencies of floods were given in revised manuscript. The Table1(in original manuscript) for comparison between observed and simulated return interval peak discharges for the period 1958–2012 in Yalu River were replaced by Appendix A4 in new manuscript. The picture can not only clarify why GEV was used to estimate design floods, but also explain the accuracy of the simulation results for flood frequency analysis (Line 399-415 and Line 894 in Mark-up manuscript version).

[Figure]

A4. Comparison between the observed and simulated return interval peak discharges in the Ai River and Yalu River based on the GEV and P-III methods. The design floods for the period 1958–2012 in Ai River (a) and Yalu River (b), and (c) The design floods for the period 1000-2012 in total Yalu River.

Q: Line 351 and later - 'frequencies of immense floods of 22.0%...' it is not clear to me what 22% means in this context! It is crucial that the authors clarify this as it is one of the main quantitative metrics used in the manuscript.

Response:

**Line 384-389 in revised manuscript:** The terms of 'Frequencies of immense floods of 22.0%' means the numbers of recorded immense floods per 100 years was 22. Inaccuracy of descriptions caused confusion of readers, and this section has been revised in the new manuscript (Line 451-460 in Mark-up manuscript version).

**Line: 399 in revised manuscript:** Table2 in original manuscript have been changed to Table1, and related contents were revised (Line 470 in in Mark-up manuscript version).

Q: Line 407 - 'observed' - I think 'estimated' is more appropriate.

Response: Revised (Line 532 in Mark-up manuscript version).

Q: Lines 421-422 - this seems to be a bit too specific an explanation given the model's limited anthropogenic representation.

Response: We have followed your suggestion (Line 546-547 in Mark-up manuscript version).

Q: Figure 8 and associated text - the figure needs to be better explained. It is not at all clear what it is showing.

Response:

**Line 515 in revised manuscript:** This section and caption of this figure have been revised in new manuscript (Line 605-609 in Mark-up manuscript version).

Q: Figure 9 and associated text - the figure is not immediately clear and could benefit from an explanation on how to interpret it. The results drawn from it are not at all apparent e.g. line 473, 479 & 487.

Response:

**Line 551-565 and line 531, 539 & 546 in revised manuscript:** This figure and associated caption were revised and Table 2 was also added in response to review's suggestion (Line 646-660 and Line 626,634 & 641 in Mark-up manuscript version).

Q: Line 500 - 'flood magnitudes' - Figure 10 title is % frequency of floods - is it magnitude or frequency?

Response:

In the original manuscript, the ratio of the number of simulated flood peaks exceeding the standards to the statistical years is defined as 'frequency of flood occurrence', and designed floods of different recurrence intervals estimated by simulated annual peak discharges during 1000-2012 were applied for the standards. Distinguishing frequency of floods and 'frequency of flood occurrence' easily make reader confusions. Actually, the impacts of climate change and human activities on floods are able to clarify through discussing the changes of frequencies of floods or magnitudes of design floods. Therefore, we removed the Sections related to 'frequency of flood occurrence' to dispel confusions and redundancies (Line 661-684 in Mark-up manuscript version).

Q: Lines 538-542- is this based on the results or general assertion?

Response:

It is general assertion based on previous researches (a change in global or regional climate patterns;

hydrological characteristics of medium and small rivers) and conclusions of this paper. We would like to let the readers to know possible changing trend of floods in the future through this assertion. We are not to make conclusion without careful investigation and available data.

Q: Line 544 and elsewhere - 'coupled' may be misleading in this context because it implies that the two models are dynamically coupled where, to my understanding, the output of ECHO-G is used as input dataset to HYDROTREND (just like any other input dataset).
Response:
We have followed this suggestion in the revised manuscript. We agree that 'coupled' seems like to imply the interactions between two model systems, similar to a coupled model of waves and tides in the ocean. In this study output of ECHO-G is only used to as input dataset.

**Mark-up manuscript version:**

[revised manuscript text omitted]

Buffer zone = statistic years / 3

lowest rainfall          location of average          peak rainfall

dry years          average years          wet years

Buffer zone

Ranked yearly precipatation during 1958-2012 (55 years)

890 A4. Comparison between the observed and simulated return interval peak discharges in the Ai River and Yalu River based on the GEV and P-III methods. The design floods for the period 1958–2012 in Ai River (a) and Yalu River (b), and (c) The design floods for the period 1000-2012 in total Yalu River.

---

## Author Response (AR2)

**Response to Editor and Review1**

Comments: Thank you for the revised version, where you have satisfactorily added the information we asked for; also the referee (the one who suggested more changes to your first submission) is satisfied with the content of the revised manuscript, but suggests to carefully revise the English.

Response:

The manuscript has been carefully revised for language, grammar, and improved clarity in response to review's comments. Mark-up revised manuscript is listed as below.

**Mark-up manuscript version:**

[revised manuscript text omitted]